# FROM SEARCH TO SAMPLING: GENERATIVE MODELS FOR ROBUST ALGORITHMIC RECOURSE

**Prateek Garg**[*]   **Lokesh Nagalapatti**   **Sunita Sarawagi**
Indian Institute of Technology Bombay

## ABSTRACT

Algorithmic Recourse provides recommendations to individuals who are adversely impacted by automated model decisions, on how to alter their profiles to achieve a favorable outcome. Effective recourse methods must balance three conflicting goals: proximity to the original profile to minimize cost, plausibility for realistic recourse, and validity to ensure the desired outcome. We show that existing methods train for these objectives separately and then search for recourse through a joint optimization over the recourse goals during inference, leading to poor recourse recommendations. We introduce GenRe, a generative recourse model designed to train the three recourse objectives jointly. Training such generative models is non-trivial due to lack of direct recourse supervision. We propose efficient ways to synthesize such supervision and further show that GenRe's training leads to a consistent estimator. Unlike most prior methods, that employ non-robust gradient descent based search during inference, GenRe simply performs a forward sampling over the generative model to produce minimum cost recourse, leading to superior performance across multiple metrics. We also demonstrate GenRe provides the best trade-off between cost, plausibility and validity, compared to state-of-art baselines. Our code is available at: https://github.com/prateekgargx/genre.

## 1 INTRODUCTION

Machine learning models are increasingly used in high-stakes decision-making areas such as in finance (Josyula et al., 2024), judiciary (Elyounes, 2019), healthcare (Burger, 2020), and hiring (Schumann et al., 2020), prompting the need for transparency and fairness in decision-making (Barocas et al., 2023). This has driven the development of tools and techniques to provide *recourse*, redress mechanisms for individuals adversely impacted by these model decisions, a legal requirement in certain jurisdictions (Kaminski, 2019). For example, consider a loan applicant with profile $x$ who is denied loan. Recourse seeks to answer the question, *'How should $x$ improve their profile to an alternative $x^+$ to secure a loan in the future?'*. A good recourse instance $x^+$ should be (1) valid, i.e., it achieves the desired label, (2) proximal to $x$ to minimize the effort involved in implementing recourse, and (3) be a plausible member of the desired class. The decision-making model is often proprietary and is kept hidden to safeguard enterprises from strategic gaming (Hardt et al., 2016), model theft (Reith et al., 2019), and other risks. We are given instead examples of individuals who got the desired label (positive instances), and those who did not (negative instances). Our goal is to learn a recourse mechanism using just these samples.

The recourse problem has been widely explored under frameworks like algorithmic recourse (Ustun et al., 2019), counterfactual explanations (Wachter et al., 2017), and contrastive explanations (Karimi et al., 2022). Most prior work assume direct access to the decision-making model, and search for the recourse instance through constrained optimization during inference on objectives which encourage low cost and validity (Karimi et al., 2022; Wachter et al., 2017; Mothilal et al., 2020; Laugel et al., 2017). Others aim for robust recourse by searching for an $x^+$ that remains valid under small changes to either the model or to the proposed $x^+$ itself (Pawelczyk et al., 2020b; Rawal et al., 2020). The plausibility criterion is often overlooked, and some methods propose to address it by training generative models (Joshi et al., 2019; Pawelczyk et al., 2020a; Downs et al., 2020). Most recently, Friedbaum et al. (2024) attempts to increase the validity of $x^+$ by including a separate verifier model that checks whether $(x, x^+)$ belong to different classes after the optimizer generates a candidate $x^+$. One significant limitation of all prior methods is that none of them are *trained* to jointly optimize the three conflicting recourse criteria of validity, proximity, and plausibility. Instead, during inference,

---

[*]Correspondence to: Prateek Garg <prateekg@iitb.ac.in>

they either ignore some recourse objectives or rely on non-robust gradient-based search on the joint objective to balance the trade-offs between them. We show in Figure 2 that this disconnect between training and inference leads to poor recourse outputs.

In a significant departure from the existing paradigms, we take a generative approach to recourse. We train a novel recourse likelihood model $\mathcal{R}_\theta(\boldsymbol{x}^+|\boldsymbol{x})$, that when conditioned on any instance $\boldsymbol{x}$ seeking recourse, outputs a distribution over likely recourse instances. Using such a model during inference, we just forward sample recourse instances, unlike existing methods that perform gradient descent optimization to find one $\boldsymbol{x}^+$. However, to train such a model we would need several instance pairs $\{(\boldsymbol{x}_i, \boldsymbol{x}'_i) : i = 1 \ldots N)\}$ where $\boldsymbol{x}'_i$ is a good recourse for any $\boldsymbol{x}_i$ with an unfavorable outcome. Such direct supervision is lacking. We show how to leverage standard unpaired classification data to sample a training set of instance pairs. We theoretically show that the sampled pairs are consistent, and the error of the expected recourse instance converges at the rate $\mathcal{O}(1/N_+)$ where $N_+$ denotes the number of instances in the positive class.

In terms of empirical validation, we first present a visual demonstration of the pros and cons of various methods using three 2-D datasets. We then evaluate on three large real-life datasets that are popularly used in the recourse literature and compare our results with eight existing methods. We show that our method achieves (1) the best score combining validity, proximity and plausibility, (2) is more robust to changes in the cost magnitude compared to SOTA likelihood-based methods, and (3) more faithfully learns the conditional density of recourse than methods that separately learn unconditional data density.

## 2 PROBLEM FORMULATION

Let $\mathcal{X} \subseteq \mathbb{R}^d$ represent the input space and $\mathcal{Y} = \{0, 1\}$ the output space where the joint distribution between them is $P(X, Y)$. In the recourse task, one label, say 0 is an unfavorable outcome (e.g., loan rejection) and the other label $y^+ = 1$ denotes a favorable one (e.g., loan approval). We are given a training dataset of $n$ instances $\mathcal{D} = \{(\boldsymbol{x}_i, y_i)\}_{i=1}^n$ sampled i.i.d. from $P(X, Y)$. We use $\mathcal{D}_1$ and $\mathcal{D}_0$ to denote the subset of $\mathcal{D}$ with label 1 and 0 respectively. We assume that we have a pretrained classifier $h : \mathcal{X} \mapsto [0, 1]$ trained on the available dataset $\mathcal{D}$, serves as an approximation for $P(Y = 1|X)$. Given an instance $\boldsymbol{x} \sim P(X)$, recourse is sought on instances where $h(\boldsymbol{x}) < 0.5$. Our objective is to design a mechanism, $\psi : \mathcal{X} \mapsto \mathcal{X}$, that outputs a recourse instance $\boldsymbol{x}^+$ such that $P(y^+|\boldsymbol{x}^+) > 0.5$, while minimizing the cost of shifting $\boldsymbol{x}^+$ measured in terms of a cost function $C : \mathcal{X} \times \mathcal{X} \mapsto \mathbb{R}^+$. $\ell_1$ distance is a popular choice for $C$. Additionally, the recourse instance $\boldsymbol{x}^+$ should be representative in the desired class distribution $P(X|y^+)$, and not an outlier. In real-life applications, weird unrepresentative profiles may not be achievable, and mislead the purpose of recourse. For instance, in domains such as banking, ensuring high $P(\boldsymbol{x}^+|y^+)$ is important because an user should not be gaming the system to get loan approval.

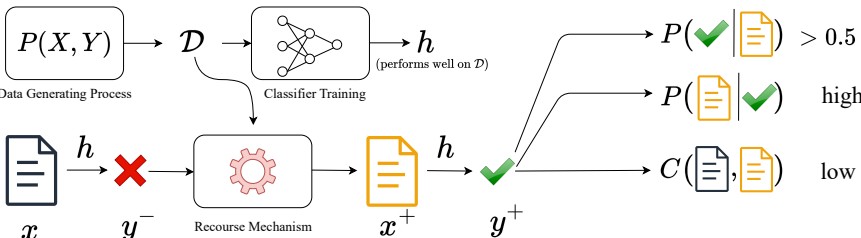

Figure 1: The recourse pipeline starts with any instance $\boldsymbol{x}$ that received an unfavorable label $h(\boldsymbol{x}) = y^-$. The recourse mechanism outputs an alternative $\boldsymbol{x}^+$ such that $h(\boldsymbol{x}^+) = y^+$. The user is satisfied as long as $\boldsymbol{x}^+$ (1) is *valid* i.e., achieves the desired label from $P(y^+|\boldsymbol{x}^+)$, (2) is *plausible* and actionable in real-life, and (3) is *proximal* to the original $\boldsymbol{x}$ to incur low cost.

Based on the above considerations an ideal recourse mechanism can be defined as follows:
$$\psi(\boldsymbol{x}) = \arg\min_{\boldsymbol{x}^+} \lambda C(\boldsymbol{x}, \boldsymbol{x}^+) - \log P(\boldsymbol{x}^+|y^+) \text{ s.t. } P(y^+|\boldsymbol{x}^+) > 0.5 \tag{1}$$

where $\lambda$ is a balance parameter which helps to trade-off cost with plausibility. Additionally, not all features can be altered for $\boldsymbol{x}$ – for example, in loan applications, a recourse mechanism should not suggest recourse where immutable attributes like race are different. We assume that the cost function models immutability and for any two instances $\boldsymbol{x}, \boldsymbol{x}'$ where immutable attributes are different,

$C(\boldsymbol{x}, \boldsymbol{x}') \to \infty$. For $\boldsymbol{x}, \boldsymbol{x}'$ where immutable attributes are same, we use $\ell_1$ distance as cost as suggested in Wachter et al. (2017).

## 3 RELATED WORK

We will illustrate the working of various methods using three 2D binary classification datasets as shown in Figure 2. Training instances are shown in light red (for $\mathcal{D}_0$) and blue color (for $\mathcal{D}_1$). Recourse is sought on instances $\boldsymbol{x}$ marked in dark red color and they are connected by an edge to the corresponding recourse instance returned by various methods. Experimental details of these figures appear in Section 5.

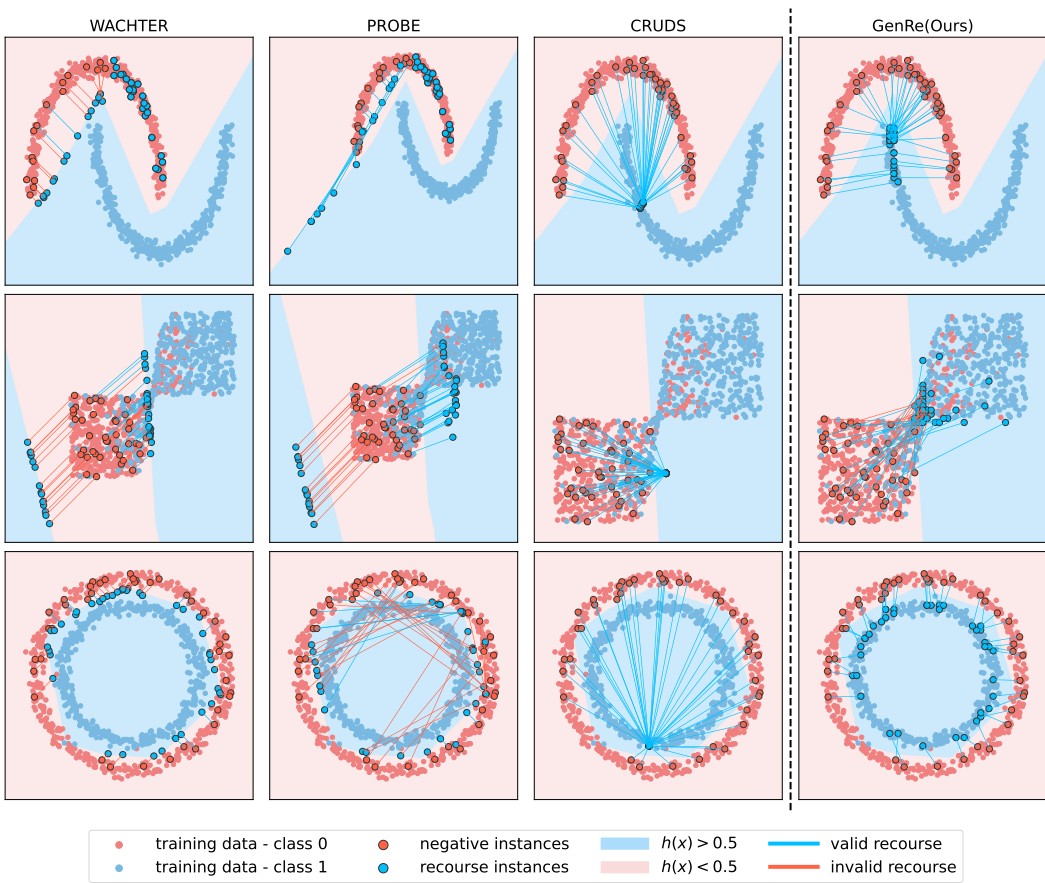

Figure 2: Comparison of different classes of recourse methods. Training instances are shown in light red and blue colors. Recourse is sought on instances marked in dark red color and they are connected by an edge to the recourse instance they were mapped to. From left to right:**(1)** Wachter, a cost minimizing method maps instances to classifier boundaries away from the blue data distribution.**(2)** PROBE, a robust recourse method, maps instances away from the boundary and from the blue data distribution. **(3)** CRUDS, a likelihood based method suffers mode collapse and for the circles dataset strays away from data distribution. to train **(4)** GenRe(our method) produces plausible recourse instances by being on the blue cloud while also minimizing cost. Recourse instances are also diverse.

We group prior work into three categories based on how they approach the recourse problem.

**Cost Minimizing Methods.** This class of methods search for instances that minimize cost under the constraint that classifier $h$ assigns the desired label, that is, $h(x) > 0.5$. The search is performed in various ways: GS (Laugel et al., 2017) uses a random search, whereas Wachter (Wachter et al., 2017) uses gradient search on a differentiable objective based on cost and classifier $h$ as follows:

$$\psi(\boldsymbol{x}) = \underset{\boldsymbol{x}^+}{\arg\min}\ C(\boldsymbol{x}, \boldsymbol{x}^+) - \gamma \log(h(\boldsymbol{x}^+)) \tag{2}$$

`DICE` (Mothilal et al., 2020) extends the above objective to include diversity. As noted by Fokkema et al. (2024), such methods often push recourse instances near the boundary of $h$, as can be seen in Figure 2 (first column). In particular, these methods suffer because: **a.)** Recourse instances that lie on boundary of $h$ have equal probability of being assigned either label and therefore are prone to be invalidated. **b.)** Recourse instances can be away from data manifold and thus unrealistic and unreliable since these methods do not consider data density.

Next we discuss two classes of methods which attempt to improve on these shortcomings.

**Robust Recourse Methods.** This class of methods trade-off cost in order to generate recourse instances which are more *robust* by generating recourse instances at some 'margin' away from the boundary of $h$. `ROAR` (Upadhyay et al., 2021) attempts to output recourse instances which are robust to small changes to the classifier $h$ by optimising for worst case model shift under the assumption that the change in parameters of $h$ is bounded. `PROBE` (Pawelczyk et al., 2023) attempts to generate instances which are robust to small perturbations to the recourse instances. These methods also do not consider data density and thus are prone to generate recourse instances which are unrealistic and unreliable as shown in Figure 2 (middle column) and may land in low density regions and/or around spurious decision boundaries. Recently, `TAP` (Friedbaum et al., 2024) introduces a verifier model which checks whether two given instances belong to the same class. This verifier is used to check if a negative-recourse instance pair belong to different class.

**Plausibility Seeking Methods.** Methods in this class leverage generative models to ensure that they predict plausible recourse by staying close to the training data manifold. `REVISE` (Joshi et al., 2019), `CRUDS` (Downs et al., 2020), and `CCHVAE` (Pawelczyk et al., 2020a) train variants of VAE (Kingma & Welling, 2014) on the training data, and then during inference they either do a gradient search on latent space, or perform rejection sampling on forward samples generated by the VAE. Suppose $D_\theta : \mathcal{Z} \rightarrow \mathcal{X}$ denotes a VAE decoder, their recourse objective during inference is:

$$\psi(\boldsymbol{x}) = D_\theta \left( \arg\min_z -\log(h(D_\theta(z))) + \lambda \cdot C(\boldsymbol{x}, D_\theta(z)) \right) \tag{3}$$

Gradient Search for $z$ on the above objective fails to keep the generated recourse example within the data distribution and suffers from mode collapse as we show in Figure 2 (Third column). Notice that particularly for the circle dataset, the recourse instances are not at all within the data distribution, and collapse to a single point.

We provide a more extensive overview of related work in Appendix A, and present comparisons on real-life data in Section 5.

## 4 OUR APPROACH

The above discussion highlights that the main challenge in recourse is jointly optimizing three conflicting objectives. The cost function $C(\boldsymbol{x}, \boldsymbol{x}^+)$ is minimum near $\boldsymbol{x}$ where the density $P(\boldsymbol{x}^+|y^+)$ of the desired positive class is low. The learned classifier $h(\boldsymbol{x}^+)$ could be maximized at regions where positive class density is low, particularly in the presence of spurious boundaries. The regions of high positive density could be far away from $\boldsymbol{x}$ leading to high costs. Unlike all prior methods that optimize for these terms during inference, we train a recourse likelihood model $\mathcal{R}_\theta(\boldsymbol{x}^+|\boldsymbol{x})$ that when conditioned on an input negative instance provide a distribution over possible recourse instances $\boldsymbol{x}^+$. We start by rewriting the ideal recourse objective defined in Equation 1 as

$$\psi(\boldsymbol{x}) = \arg\max_{\boldsymbol{x}^+} \exp\left(-\lambda C(\boldsymbol{x}, \boldsymbol{x}^+)\right) P(\boldsymbol{x}^+|y^+) V(\boldsymbol{x}^+) \tag{4}$$

where $V(\boldsymbol{x}^+) = \delta(P(y^+|\boldsymbol{x}^+) > 0.5)$ denotes the desired validity constraint on a recourse instance. Using the above we define the ideal un-normalized recourse likelihood as:

$$R(\boldsymbol{x}^+|\boldsymbol{x}) \propto \exp\left(-\lambda C(\boldsymbol{x}, \boldsymbol{x}^+)\right) P(\boldsymbol{x}^+|y^+) V(\boldsymbol{x}^+). \tag{5}$$

Our goal during training is to learn a model $\mathcal{R}_\theta$ to estimate this ideal recourse density $R(\boldsymbol{x}^+|\boldsymbol{x})$. The main challenge here is that we are given only individual labeled instances $\mathcal{D} = \mathcal{D}_1 \cup \mathcal{D}_0$, whereas what we ideally want is instance-pair sampled from $R(\boldsymbol{x}^+|\boldsymbol{x})$. In Section 4.1 we present how we construct such samples from $\mathcal{D}$, and then present how we use the constructed pairs to architect and train our estimated model $\mathcal{R}_\theta(\boldsymbol{x}^+|\boldsymbol{x})$. In Section 4.2, we describe one convenient model parameterization which allows for sampling.

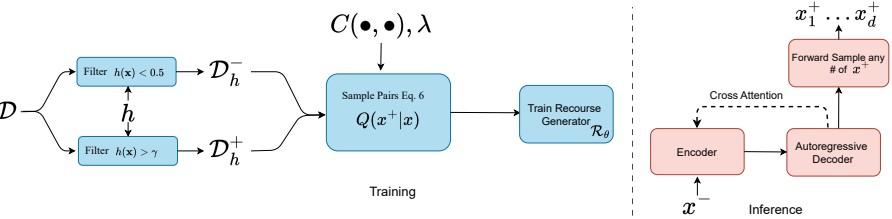

Figure 3: Overview of GenRe. We define an empirical distribution of instance pairs $(\boldsymbol{x}, \boldsymbol{x}^+)$ using training data $\mathcal{D}$, classifier $h$, cost function $C$ and balance parameter $\lambda$ to train the recourse model $\mathcal{R}_\theta$, an encoder-decoder model. During inference, the given negative instance $\boldsymbol{x}$ is fed to the decoder, and recourse instances sampled from the decoder auto-regressively.

## 4.1 Creating Training Pairs

We define $\mathcal{D}_h^- := \{\boldsymbol{x}_i \in \mathcal{D} | y_i = 0, h(x_i) \le 0.5\}$. These denote training instances where recourse is desired. Similarly define $\mathcal{D}_h^+ := \{\boldsymbol{x}_i \in \mathcal{D} | y_i = y^+, h(x_i) > \gamma\}$ for some chosen $\gamma \ge 0.5$. These denote the subset of training examples from the positive class that are also predicted positive by the classifier with confidence $\gamma$.

For each $\boldsymbol{x} \in \mathcal{D}_h^-$, we define an empirical distribution as follows:

$$Q(\boldsymbol{x}^+|\boldsymbol{x}) = \begin{cases} \frac{e^{-\lambda C(\boldsymbol{x}, \boldsymbol{x}^+)}}{\sum_{\boldsymbol{x}^+ \in \mathcal{D}_h^+} e^{-\lambda C(\boldsymbol{x}, \boldsymbol{x}^+)}} & \text{if } \boldsymbol{x}^+ \in \mathcal{D}_h^+ \\ 0 & \text{otherwise} \end{cases} \tag{6}$$

where $\lambda$ denotes the balance parameter as described in 1. We show in Section 4.4 that $Q(\boldsymbol{x}^+|\boldsymbol{x})$ is a consistent estimator of $R(\boldsymbol{x}^+|\boldsymbol{x})$, and the difference in the expected value of the recourse instance $\|\mathbb{E}_Q[\boldsymbol{x}^+|\boldsymbol{x}] - \mathbb{E}_R[\boldsymbol{x}^+|\boldsymbol{x}]\|$ reduces at the rate of $\frac{1}{N_+}$, where $N_+ = |\mathcal{D}_h^+|$ denotes the number of positive instances, when $h(\boldsymbol{x})$ is the actual conditional distribution $P(Y|X)$. We note that as $\lambda \to \infty$, $Q$ will always return the nearest neighbor of $\boldsymbol{x}$ in $\mathcal{D}_h^+$. In section 5.4, we compare our method GenRe with nearest neighbor search from $\mathcal{D}_h^+$.

## 4.2 Architecture and Training of the Generative Model

We choose $\mathcal{R}_\theta$ to be an auto-regressive generative model so that during inference we only need to perform a forward sampling on the model to obtain recourse instances. We next describe the architecture of $\mathcal{R}_\theta$. We use a transformer-based encoder-decoder architecture. The negative instance $\boldsymbol{x}$ is input to the encoder with learned position embeddings. The decoder defines the output recourse distribution auto-regressively as $\mathcal{R}_\theta(\boldsymbol{x}^+|\boldsymbol{x}) = \prod_{j=1}^d \mathcal{R}_\theta(x_j^+|\boldsymbol{x}, x_1^+, \ldots x_{j-1}^+)$. The decoder also uses learned position embeddings and performs cross attention on the encoder states and causal self-attention on decoder states. The output conditional distribution for each attribute $x_j^+$ is modeled as a kernel density:

$$\mathcal{R}_\theta(x_j^+|\boldsymbol{x}, x_1^+, \ldots x_{j-1}^+) = \sum_{k=1}^{n_j} p_{j,k}^\theta \cdot \mathcal{K}((x_j^+ - \mu_{j,k})/w_{j,k})$$

Here, $p_{j,k}^\theta \ge 0$, such that $\sum_k p_{j,k}^\theta = 1$, represents the kernel weights output by the transformer, and is implemented as a Softmax layer. The means $\mu_{j,k}$ and width $w_{j,k}$ of the $k^{\text{th}}$ component are fixed by binning the $j^{\text{th}}$ attribute in $\mathcal{D}_h^+$ into $n_j$ partitions, with $\mu_{j,k}$ as the bin center and $w_{j,k}$ as the bin width. We use the RBF Kernel. The loss for a paired sample $(\boldsymbol{x}^+, \boldsymbol{x}) \sim Q$ is computed as,

$$\mathcal{L}(\boldsymbol{x}^+|\boldsymbol{x}; \theta) = -\sum_{j=1}^d \log \left( \sum_{k=1}^{n_j} p_{j,k}^\theta \cdot \mathcal{K}((x_j^+ - \mu_{j,k})/w_{j,k}) \right) \tag{7}$$

$$\le -\sum_{j=1}^d \sum_{k=1}^{n_j} \left( \frac{\mathcal{K}((x_j^+ - \mu_{j,k})/w_{j,k})}{\sum_{l=1}^{n_j} \mathcal{K}((x_l^+ - \mu_{l,k})/w_{l,k})} \right) \cdot \log p_{j,k}^\theta + C \tag{8}$$

Here we have used Jensen's inequality to express the loss as a simple cross entropy loss on the $p_j^\theta$ vector output by the last softmax layer with the kernel ratios serving as soft-labels.

The overall training process of the model is shown in Figure 3 and outlined in Algorithm 1. We emphasize that this is one particular choice of parameterisation, with a note that other models can also be considered. In appendix 5.6, we show results on pre-trained diffusion models with guidance to sample from distribution described in 5.

### 4.3 INFERENCE

Once the auto-regressive encoder-decoder is trained, finding recourse on any input negative instance $x$ during test time, entails just a simple forward sampling step on $\mathcal{R}_\theta$. We input $x$ to the encoder, and sample the recourse instance auto-regressively feature-by-feature. The full inference method appears in Algorithm 2. Another advantage of our approach is that a user can sample multiple recourse instances and choose which recourse action to implement.

### 4.4 THEORETICAL ANALYSIS

**Theorem 4.1.** *Let $f(x^+, x)$ be any function of $(x^+, x)$. For $Q(x^+|x)$ defined in Equation 6 and $R(x^+|x)$ defined in Equation 4, let $\mu = \mathbb{E}_R[f]$ and $\hat{\mu} = \mathbb{E}_Q[f]$. Then $\hat{\mu}$ is a consistent estimate of $\mu$ when $h(x) > 0.5$ and $P(y^+|x) > 0.5$ agree.*

*Proof.* Let $D_1$ denote the subset of $\mathcal{D}$ where $y_i = y^+$. Substituting the definition of $Q$, it is easy to see that

$$\mathbb{E}_Q[f(\bullet|x)] = \sum_{(x^+,1) \in D_1} f(x^+, x) \frac{e^{-\lambda C(x,x^+)} V(x^+)}{\sum_{x^+ \in \mathcal{D}_h^+} e^{-\lambda C(x,x^+)} V(x^+)} \tag{9}$$

if $V(x^+) = \delta(P(y^+|x) > 0.5) = \delta(h(x) > 0.5)$. The above can be seen as a normalized importance weighted estimator for $\mu = \mathbb{E}_R[f]$ when we treat $P(x^+|y^+)$ as a proposal distribution for $R(x^+|x)$ distribution. The proposal distribution is sound since $P(x^+|y^+) > 0$ whenever $R(x^+|x) > 0$ because $R$ includes $P(x^+|y^+)$ as one of its terms. The training data $D_1$ is a representative sample from $P(X|y^+)$. The unnormalized weight of the importance sampling step $\tilde{w}(x^+) = \frac{e^{-\lambda C(x,x^+)} V(x^+) P(x^+|y^+)}{P(X|y^+)} = e^{-\lambda C(x,x^+)} V(x^+)$. Substituting these in Eq 9 we see that $\mathbb{E}_Q[f]$ is a self-normalized importance weighted estimate, which is well-known to be a consistent estimator when proposal is non-zero at support point of target. $\square$

**Theorem 4.2.** *The difference in the expected value of the counterfactual $\|\mathbb{E}_Q[x^+|x] - \mathbb{E}_R[x^+|x]\|$ reduces at the rate of $\frac{1}{N_+}$ when $h(x)$ is the actual conditional distribution $P(Y|X)$.*

*Proof.* Using $f(x^+, x) = x^+$, the proof of Theorem 4.1 showed that $\mathbb{E}_Q[x^+|x]$ is a self-normalized importance sampling estimate of $\mathbb{E}_R[x^+|x]$. The variance of this estimate for a given $x$ is approximately $\frac{1}{|D_1|} \text{Var}_{R(X|x)}[f(X, x)](1 + \text{Var}_{P(X|y^+)}[e^{-\lambda C(x,X)} V(X)])$ which reduces at the rate $\frac{1}{|D_1|}$ (Koller & Friedman (2009), Chapter 12). $\square$

## 5 EXPERIMENTAL RESULTS

We next present empirical comparisons of our generative recourse model with several prior recourse methods. We already presented a visual comparison on synthetic datasets in Figure 2. In this section we focus on real datasets.

### 5.1 EXPERIMENTAL SETUP

**Datasets:** We experimented with benchmark datasets commonly used to evaluate recourse algorithms: Adult Income (Becker & Kohavi, 1996), FICO HELOC (FICO, 2018), and COMPAS (Angwin et al., 2016). These datasets contain a mix of continuous and categorical features. All continuous attributes are normalized to the range of $[0, 1]$, while categorical attributes are one-hot encoded. We defer detailed descriptions of these datasets to Appendix C.1 and present a summary in Table 1. For each dataset, we train a Random Forest (RF) classifier to mimic the *latent* decision-making model, which assigns the gold labels. No method has access to this classifier during training; instead, they have access to training data $\mathcal{D}$ sampled from it. This classifier is used to evaluate the

validity of recourse output by various methods. We ensure that the RF classifier is calibrated by using the `CalibratedClassifierCV` API from `sklearn` (Pedregosa et al., 2011).

**Baselines:** We compare our method with eight prior recourse methods covering each of the three category of prior methods already described in Section 3: `Wachter` (Wachter et al., 2017), `GS` (Laugel et al., 2017), `DICE` (Mothilal et al., 2020), `ROAR` (Upadhyay et al., 2021), `PROBE` (Pawelczyk et al., 2023), `REVISE` (Joshi et al., 2019), `CRUDS` (Downs et al., 2020), and `CCHVAE` (Pawelczyk et al., 2020a). `TAP` (Friedbaum et al., 2024). For standardized comparison, we used their public implementation from CARLA recourse library[1] (Pawelczyk et al., 2021).

**Implementation Details.** For the labeled dataset $\mathcal{D}$, we adopt the features from the real data as is and assign labels sampled from the RF classifier. The classifier $h(x)$ is an Artificial Neural Network (ANN) – a ReLU-based model with three hidden layers of size 10 each, trained with a learning rate of 0.001 for 100 epochs using a batch size of 64. The accuracy of $h(x)$ is reported in Table 1.

| Dataset | #Feat. | #Cat. | #Immut. | #Pos | #Neg | #Train | #Test | ANN accuracy |
|---|---|---|---|---|---|---|---|---|
| Adult Income | 13 | 7 | 2 | 8,742 | 27,877 | 36624 | 12208 | 77.33 |
| COMPAS | 7 | 4 | 2 | 3,764 | 865 | 4629 | 1543 | 69.60 |
| HELOC | 21 | 0 | 0 | 3,548 | 3,855 | 7403 | 2468 | 74.23 |

Table 1: Data Statistics along with accuracy of ANN classifier

For training $\mathcal{R}_\theta$, we use a Transformer (Vaswani et al., 2017) from PyTorch (Paszke et al., 2019) with learned position embedding, embedding size 32, and 16 layers in each of encoder and decoder, and 8 heads. The number of bins in the last layer is 50. We choose the value of $\lambda = 5.0$ when sampling training pairs. During inference (Algorithm 2), we set the temperature for bin selection $\tau = 10.00$ and $\sigma = 0.00$, generate 10 samples and choose the sample which gets highest probability from the classifier $h(x)$. In Appendix D.2, we provide results over other values of $\tau$ and $\sigma$. We describe other relevant hyperparameters in Appendix C.2.

**Performance Metrics.** We evaluate the performance of a recourse method on a test set $\{x_i\}_{i=1}^m$ consisting of $m$ negative instances using the following metrics:

1. *Cost*: We define cost as the $\ell_1$ distance between the negative instance $x$ and its corresponding recourse instance $x^+$.
2. *Val*: Validity measures if the recourse was successful, that is, the RF classifier assigns $y^+$.
3. *LOF*: LOF(Breunig et al., 2000) assesses how plausible or representative the recourse instance is. We report the fraction of recourse instances which were assigned as inliers by this module.
4. *Score*: To evaluate all methods using a single metric, we define *Score* as *Score = Val + LOF -* $\frac{Cost}{d}$, where $d$ is the number of features in the dataset. Note that the maximum possible value of *Score* is 2.

| **Dataset** | **Adult Income** | | | | **COMPAS** | | | | **HELOC** | | | |
|---|---|---|---|---|---|---|---|---|---|---|---|---|
| **Method** | Cost ↓ | Val ↑ | LOF ↑ | Score ↑ | Cost ↓ | Val ↑ | LOF ↑ | Score ↑ | Cost ↓ | Val ↑ | LOF ↑ | Score ↑ |
| Wachter | 0.31 | 0.51 | 0.76 | 1.24 | 0.20 | 0.59 | 0.23 | 0.79 | 0.79 | 0.35 | 0.97 | 1.28 |
| GS | 1.82 | 0.40 | 0.49 | 0.75 | 1.20 | 0.66 | 0.48 | 0.97 | 0.58 | 0.32 | 0.94 | 1.23 |
| DICE | 0.22 | 0.62 | 0.73 | 1.33 | 0.19 | 0.57 | 0.44 | 0.99 | 0.49 | 0.34 | 0.97 | 1.29 |
| ROAR | 10.17 | 0.96 | 0.01 | 0.19 | 4.01 | 0.87 | 0.01 | 0.30 | 9.64 | 0.47 | 0.17 | 0.19 |
| PROBE | 33.91 | 1.00 | 0.00 | -1.61 | 5.77 | 0.82 | 0.00 | -0.00 | 5.37 | 0.69 | 0.07 | 0.49 |
| TAP | 1.10 | 0.99 | 0.67 | 1.57 | 1.46 | 0.88 | 0.70 | 1.37 | 0.71 | 0.36 | 0.53 | 0.86 |
| CCHVAE | 2.11 | 0.00 | 1.00 | 0.84 | 3.03 | 1.00 | 0.07 | 0.64 | 3.58 | 0.04 | 0.14 | 0.02 |
| CRUDS | 3.17 | 1.00 | 0.96 | 1.72 | 1.10 | 0.98 | 1.00 | 1.83 | 4.30 | 1.00 | 0.57 | 1.37 |
| GenRe | 0.69 | 1.00 | 0.98 | **1.93** | 0.51 | 0.99 | 0.97 | **1.89** | 2.01 | 1.00 | 1.00 | **1.90** |

Table 2: Comparing different recourse mechanisms on three real-life datasets. For each mechanism, we report *cost*, *validity*, *LOF*, and a combined *Score=Val+LOF-Cost/d*. GenRe provides the best score across all datasets, and is close to 2, the maximum achievable score.

---
[1] github.com/carla-recourse/CARLA

## 5.2 Overall comparison with baselines

Table 2 presents a comparison of our method GenRe with several other baselines on the cost, validity, and LOF metrics, and an overall combined score. From this table we can make a number of important observations: (1) First, observe that our method GenRe consistently provides competent performance across all metrics on all three datasets. GenRe's average score across the three datasets is the highest. (2) If we compare on cost alone, the first three methods (Wachter, GS and DICE) that ignore plausibility, achieve lower cost than GenRe but they provide poor validity (close to 0.5). This is possibly because they optimize for validity on the learned $h(x)$ which may differ from the gold classifier. (3) The next two methods (ROAR and PROBE) choose more robust instances with a margin, and thus they achieve much higher validity. However, they struggle with choosing good margins, and the recourse instances they return are very far from $x$ as seen by the abnormally high costs. Also, these methods mostly yield outliers for recourse as seen in the low LOF scores. TAP, which trains an auxiliary verifier model to select highly valid recourse instances during inference, outperforms ROAR and PROBE, achieving good validity scores in Adult and COMPAS. However, it falls short in other recourse metrics, resulting in a consistent subpar overall score. (4) The next two methods CCHVAE and CRUDS incorporate plausibility with a VAE, and of these CRUDS provides competitive LOF and validity values but it generally incurs higher cost than our method.

## 5.3 Generating recourse from a recourse likelihood model Vs Optimizing jointly during inference.

We present a more in-depth comparison of GenRe, our training-based recourse method, with CRUDS that emerged as the second best in the comparisons above. CRUDS includes all three objectives of validity, proximity, and plausibility like in GenRe but combines them during inference. In the CRUDS implementation[2], inference entails solving an objective like in Equation 3 but with some set of $z$ fixed. Thus, $\lambda$ in this method also allows control of the magnitude of cost like in GenRe's objective (Equation 4). No matter what the $\lambda$, an actionable recourse method should always generate plausible instances from the data manifold in order to be taken seriously in real-life. The $\lambda$ can be used to tradeoff expense Vs guarantee of achieving the target label. From this perspective we compare how CRUDS and GenRe respond to changing cost magnitude $\lambda$. We perform experiments with $\lambda \in \{0.5, 1.0, 2.5, 5.0, 10.0\}$. In the Figure below we plot the value of $-\text{cost}(x, x^+)/d$ on the X-axis against validity in the first row and against LOF in the second row. Each point denotes a specific $\lambda$. From these graphs we can make the following interesting observations:

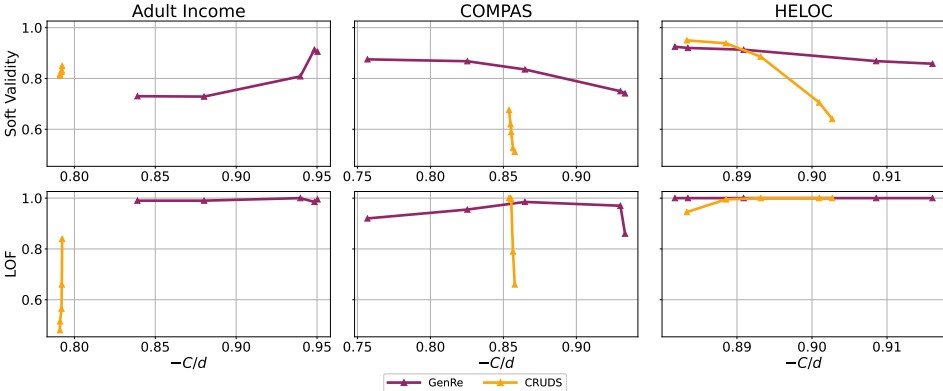

Figure 4: Comparing GenRe with CRUDS for different values of balance parameter $\lambda \in \{0.5, 1.0, 2.5, 5.0, 10.0\}$. Note that $x$-axis is on exponential scale. Top: Comparing soft validity. Bottom: Comparing fraction of recourse instances that were inliers. GenRe provides better tradeoffs than CRUDS with changing cost: GenRe always returns plausible instances and tradesoff validity gradually with cost. CRUDS shows huge swings in validity and plausibility with changing $\lambda$.

(1) On all datasets we observe that GenRe consistently provides plausible recourse instances as seen by the high LOF scores across all $\lambda$ values, even while cost increases. In contrast, except for the

---

[2]The inference method described in the CRUDS paper differs from their implementation in the CARLA library

HELOC dataset, the LOF scores of CRUDS swings significantly with changing $\lambda$ even though the cost stays the same. (2) The validity values change gradually with changing $\lambda$ whereas in CRUDS on two of the datasets COMPAS and HELOC, we observe much greater swings.

## 5.4  ABLATION: ROLE OF CONDITIONAL LIKELIHOOD

Apart from the joint training of recourse likelihood, another explanation for the superior performance of GenRe is that our approach involves training the *conditional* likelihood of recourse instances $R(x^+|x)$. The conditioning on input negative instances results in a density that is more tractable to learn than the unconditional data density of $P(X|y^+)$ that needs to capture the entire data manifold of positive instances. We show this visually on two synthetic datasets from Figure 2. For each dataset, we train two density models: $\mathcal{R}_\theta(x^+|x)$ and $P(X|y^+)$. Both these densities are learned using transformer models. Since $P(X|y^+)$ does not need an encoder, its transformer uses twice the number of layers as the conditional one. We show the contours of these two conditional densities in Figure 5. The large red dot in the plot represents the negative instance $x$ on which the density is conditioned.

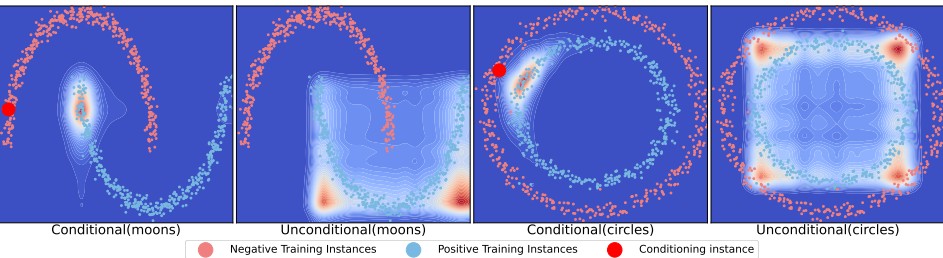

Conditional(moons)    Unconditional(moons)    Conditional(circles)    Unconditional(circles)

● Negative Training Instances    ● Positive Training Instances    ● Conditioning instance

Figure 5: Visual Comparison between contours of density learned by conditional model (odd positions) and unconditional model (even positions)

We can see vividly that the conditional density of $\mathcal{R}_\theta(x^+|x)$ is concentrated around an optimal region on the data manifold that meets all three recourse goals. In contrast, the unconditional density of the blue training points from the positive class is subpar. Such density models when combined with cost and validity objectives during inference are less likely to yield recourse on the data manifold.

## 5.5  COMPARISON WITH NEAREST NEIGHBOR SEARCH

As $\lambda \to \infty$, $Q$ (as defined in Eq. 6) will always return the nearest neighbor of $x$ in $\mathcal{D}_h^+$ as defined in Section 4.1. In this section we establish the advantages of GenRe over nearest neighbor search. We consider three variants: (a) NNR (Nearest Neighbor Recourse) that selects the closest neighbor which has the desired class label. (b) NNR ($\gamma > 0.7$) that selects recourse instances which are assigned a confidence of more than 0.7 from classifier $h$, and finally, (c) NNR ($\mathcal{D}_h^+$) that returns nearest recourse instance which in addition to having a confidence of $\gamma > 0.7$ by the classifier $h$, also enforces that observed label in the training data is $y = 1$. We present the results in table 3

| Dataset | Adult Income | | | | COMPAS | | | | HELOC | | | |
|---|---|---|---|---|---|---|---|---|---|---|---|---|
| Method | Cost ↓ | Val ↑ | LOF ↑ | Score ↑ | Cost ↓ | Val ↑ | LOF ↑ | Score ↑ | Cost ↓ | Val ↑ | LOF ↑ | Score ↑ |
| NNR | 0.36 | 0.48 | 0.89 | 1.34 | 0.17 | 0.86 | 0.88 | 1.71 | 1.22 | 0.58 | 1.00 | 1.53 |
| NNR ($\gamma = 0.7$) | 0.46 | 0.72 | 0.84 | 1.53 | 0.43 | 0.99 | 0.97 | **1.90** | 1.52 | 0.69 | 1.00 | 1.62 |
| NNR ($\mathcal{D}_h^+$) | 0.47 | 0.85 | 0.89 | 1.71 | 0.45 | 0.99 | 0.97 | **1.90** | 1.57 | 0.96 | 0.99 | 1.88 |
| GenRe | 0.69 | 1.00 | 0.98 | **1.93** | 0.51 | 0.99 | 0.97 | 1.89 | 2.01 | 1.00 | 1.00 | **1.90** |

Table 3: Comparison of Nearest Neighbor Search with GenRe. GenRe performs much better in term of LOF across all the datasets.

We make the following observations: (1) Among the NNR methods, a consistent trend emerges: NNR ($\gamma > 0.7$) consistently outperforms standard NNR, while NNR ($\mathcal{D}_h^+$) outperforms NNR ($\gamma > 0.7$) across all recourse metrics, except cost. This underscores the value of incorporating both constraints when providing recourse and further justifies the pairing approach adopted by GenRe.

(2) On the Adult Income and HELOC datasets, GenRe outperforms all variations of NNR in terms of the overall score, while achieving comparable performance on the COMPAS dataset. Note that NNR completely ignores the plausibility of instances and therefore succumbs to outliers, as reflected in their low LOF scores. GenRe, on the contrary, trades-off cost to provide instances which are more plausible. In Table 12, we also compare GenRe across a range of balance parameter $\lambda$.

## 5.6 COMPARISON WITH PRE-TRAINED DIFFUSION MODELS

Recent work on guidance in diffusion models allows for sampling from distribution of the form 5 using pre-trained models. To investigate if GenRe which trains with pairs has any advantage over such methods, we experiment with the current best performing diffusion model for tabular data, TabSyn (Zhang et al., 2024) and constrained it with a state-of-the-art derivative-free guidance method SVDD (Li et al., 2024). To ensure a fair comparison, we train the diffusion model only on $\mathcal{D}_h^+$ as described in section 4.1. We compare with two methods: (1) TabSyn+$Q$: We implement a semi-ideal, brute-force method in which we sample a dataset $\mathcal{D}_{syn}^+$, which is of the same size as original dataset. For a given negative instance $x^-$, define $Q$ on $\mathcal{D}_{syn}^+$ as described in Eq. 6. This experiments sets a skyline for guidance methods. (2) TabSyn+SVDD: SVDD requires specifying a downstream 'reward' function for conditional generation but does not require this function to be differentiable. If we let the reward function to be negative of cost, we recover the distribution in Eq. 5.

| Dataset | Adult Income | | | | COMPAS | | | | HELOC | | | |
|---|---|---|---|---|---|---|---|---|---|---|---|---|
| Method | Cost ↓ | Val ↑ | LOF ↑ | Score ↑ | Cost ↓ | Val ↑ | LOF ↑ | Score ↑ | Cost ↓ | Val ↑ | LOF ↑ | Score ↑ |
| TabSyn+$Q$ | 0.86 | 1.00 | 0.79 | 1.72 | 0.78 | 0.99 | 0.81 | 1.69 | 2.41 | 0.99 | 0.99 | 1.87 |
| TabSyn+SVDD($\lambda = 5.0$) | 3.11 | 0.99 | 0.83 | 1.58 | 2.18 | 1.00 | 0.69 | 1.37 | 3.14 | 0.98 | 0.97 | 1.8 |
| TabSyn+SVDD($\lambda = 10.0$) | 3.10 | 1.00 | 0.87 | 1.63 | 2.20 | 1.00 | 0.79 | 1.48 | 2.85 | 0.99 | 0.99 | 1.84 |
| GenRe($\lambda = 5.0$) | 0.69 | 1.00 | 0.98 | **1.93** | 0.51 | 0.99 | 0.97 | **1.89** | 2.01 | 1.00 | 1.00 | **1.90** |

Table 4: GenRe outperforms pre-trained Diffusion models with guidance

While TabSyn+$Q$ performs similarly to GenRe on the HELOC dataset, it performs worse on the Adult Income and COMPAS datasets, particularly struggling with the LOF metric. TabSyn+SVDD performs significantly worse on both datasets in terms of cost as well as the LOF metric. Additionally, we observed that in many instances, TabSyn+SVDD failed to generate recourse instances that satisfy the immutability constraints.

## 6 CONCLUSION

In this work, we proposed a model GenRe to maximize likelihood of recourse over instances that meet the three recourse goals of validity, proximity, and plausibility. We demonstrated that methods that are not trained *jointly* with the three recourse goals fail to achieve all of them during inference. We confirmed this empirically on toy 2D datasets and three standard recourse benchmarks across eight state-of-the-art recourse baselines. Another interesting property of GenRe is that it generates recourse just via forward sampling from the trained recourse model, unlike most existing methods that perform expensive and non-robust gradient descent search during inference. The main challenge we addressed in developing GenRe was training the generator given lack of direct recourse supervision. We addressed this by designing a pairing strategy that pairs each negative instance instance seeking recourse with plausible positive recourse instances in the training data. We proved these pairs are consistent, and hence GenRe's method of training on them is consistent with the ideal recourse objective. Our experiments and sensitivity analysis further demonstrate GenRe's ability to optimally balance the three recourse criteria, while remaining robust across a wide range of hyper-parameters.

## ACKNOWLEDGEMENTS

We acknowledge the support of the SBI Foundation Hub for Data Science & Analytics, and the Centre for Machine Intelligence and Data Science (C-MInDS) at the Indian Institute of Technology Bombay for providing financial support and infrastructure for conducting the research presented in this paper.

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

## A  DETAILED RELATED WORK

**Cost minimizing Methods.** Laugel et al. (2017) suggests a random search algorithm, which generates samples around the factual input point until a point with a corresponding counterfactual class label was found. The random samples are generated around $x$ using growing hyper spheres. Ustun et al. (2019) formulates a discrete optimization problem on user-specified constraints and uses integer programming solvers such as CPLEX or CBC. Karimi et al. (2020a) translates the problem as satisfiability problem which can be solved using off the shelve SMT-solvers. Wachter et al. (2017) formulates a differentiable objective using cost and classifier and generates counterfactual explanations by minimizing the objective function using gradient descent Mothilal et al. (2020) extends the differentiable objective with a diversity constraint to the objective uses gradient descent to find a solution that trades-off cost and diversity

**Robust Methods.** Several recent methods acknowledge that deployed models in practice often change with time Rawal et al. (2020), and as a consequence recourse generated using the classifier used during training can be invalidated by the deployed model. This calls for a need to *robust* recourse. Upadhyay et al. (2021) finds counterfactual in worst case model shifts under the assumption of bounded change in parameter space. Hamman et al. (2023) challenges this assumption and instead defines "natural" model shifts where model prediction do not change drastically in-distribution of training data.

Another line of work considers robustness in the case with respect to small perturbations to the input. As noted by Fokkema et al. (2024) most recourse algorithms map instances to close to the boundary of classifiers . If user implementation of recourse is noisy, the recourse can get invalidated. Pawelczyk et al. (2023) address the problem by optimising for expected label after recourse under normal noise. Recourse can also be noisy due to underlying causal model. Karimi et al. (2020b). Dominguez-Olmedo et al. (2022) suggest a minmax objective which reduces to outputting a point some 'margin' away from the boundary in the case of linear classifiers. Recent work Friedbaum et al. (2024) trains auxiliary verifier models to ensure robustness in recourse validity. Black et al. (2021) also considers recourse validity in case of a model shift and ensures that the generated counterfactuals are in smooth regions of classifier

**Likely Recourse Methods.** Joshi et al. (2019) use a VAE(Kingma & Welling (2014)) to ensure that the generated recourse instances are close to the data-manifold and perform gradient descent in latent space to optimise the recourse instance. Pawelczyk et al. (2020a) perform random search in latent space and filter out the instances which correctly flip the predictions. Downs et al. (2020) extends the method to include a class conditional VAE which can be incorporated in CCHVAE and REVISE.

Schut et al. (2021) asserts that the recourse instances should belong to the region where classification model has high certainty. They estimate the uncertainty using an ensemble of models and include that in their objective. Antorán et al. (2020) considers the case of Bayesian Neural Networks and estimate the uncertainty by sampling multiple models from the model distribution.

**Others.** More recent work includes Kanamori et al. (2024) which attempts to learn decision tree classifiers which facilitate more desirable recourse, as opposed to us where we assume that the classifiers are given to us. Gao & Lakkaraju (2024) attempts to generate recourse which is invariant across multiple subgroups. Bewley et al. (2024) attempts to generate human readable rules from a suggested recourse instances.

We note that quite a few methods (Upadhyay et al., 2021; Pawelczyk et al., 2023; Ustun et al., 2019; Dominguez-Olmedo et al., 2022) are originally developed for linear models but are extended to non-linear models like neural networks by using local linear approximation using LIME coefficients Ribeiro et al. (2016)

## B  ALGORITHMS

In this section, we specify the training and inference algorithm for the autoregressive model used in the paper.

---

**Algorithm 1** Training Recourse model $\mathcal{R}_\theta$

---

**Require:** Training data $\mathcal{D}$, lrn rate $\eta$, batch size $b$, epochs $e$, sampling parameter $K$, cost coefficient $\lambda$, pair validity $\gamma$, classifier $h$
**Ensure:** Trained model $\mathcal{R}_\theta$ with parameters $\theta$
1: $\mathcal{D}_h^- \leftarrow \{\boldsymbol{x}_i \in \mathcal{D} | y_i = 0, h(\boldsymbol{x}_i) \leq 0.5\};\ \mathcal{D}_h^+ \leftarrow \{\boldsymbol{x}_i \in \mathcal{D} | y_i = y^+, h(\boldsymbol{x}_i) > \gamma\}$
2: $\mathcal{R}_\theta \leftarrow$ Initialize $\{\mathcal{R}_\theta^{\text{encoder}}, \mathcal{R}_\theta^{\text{decoder}}\}$
3: $Q_{\text{trunc}}(\bullet | \boldsymbol{x}) = \texttt{TopK}\left(Q\left(\bullet | \boldsymbol{x}, \mathcal{D}_h^+\right), K\right)$ $\qquad \triangleright$ Truncate $Q$ (Eq. 6) to Top $K$ entries for $\boldsymbol{x} \in \mathcal{D}_h^-$
4: **for** epoch $\in [e]$ **do**
5: $\quad$ **for** each minibatch $B$ of size $b$ from $\mathcal{D}_h^-$ **do**
6: $\qquad$ Sample $\boldsymbol{x}_i' \sim Q_{\text{trunc}}(\boldsymbol{x}_i)$ for each $\boldsymbol{x}_i \in B$,
7: $\qquad$ $\mathcal{L}(\theta) =$ Sum of loss on $(\boldsymbol{x}_i, \boldsymbol{x}_i')$ using Eq. 8; $\theta \leftarrow \texttt{GradDescStep}(\mathcal{L}, \eta)$
8: $\quad$ **end for**
9: **end for**

---

**Algorithm 2** Forward Sampling one Recourse Instance

---

**Require:** negative instance $\boldsymbol{x}^-$, categorical indices $m_c$, recourse model $\mathcal{R}_\theta = \{\mathcal{R}_\theta^{\text{enc}}, \mathcal{R}_\theta^{\text{dec}}\}$, temperature $\tau$, bin means $\{\mu_{j,k}\}$, bin width $\{\sigma_{j,k}\}$
**Ensure:** predicted recourse $\boldsymbol{x}'$
1: $\boldsymbol{x}_{\text{enc}}^- \leftarrow \mathcal{R}_\theta^{\text{enc}}(\boldsymbol{x}^-)$ $\qquad\qquad\qquad\qquad\qquad \triangleright$ Encode the input neg. instance
2: $\boldsymbol{x}' \leftarrow [0, 0, \ldots 0]$ $\qquad\qquad\qquad\qquad\qquad\qquad \triangleright$ Init. $\boldsymbol{x}'$ with $d$ zeros
3: **for** $j \in [d]$ **do**
4: $\quad p_j = \texttt{SoftMax}\left(\tau \cdot \mathcal{R}_\theta^{\text{dec}}\left(\boldsymbol{x}_{\text{enc}}^-, \boldsymbol{x}_{1:j-1}'\right)\right)$ $\quad \triangleright \mathcal{R}_\theta^{\text{dec}}\left(\boldsymbol{x}_{\text{enc}}^-, \boldsymbol{x}_{1:j-1}'\right)$ is the logits over $n_j$ bins
5: $\quad$ Sample $\epsilon \sim \mathcal{N}(0, 1), k \sim p_j$
6: $\quad x_j' \leftarrow \mu_{j,k} + \sigma_{j,k} \cdot \epsilon$
7: **end for**
8: $\boldsymbol{x}' \leftarrow \texttt{projectCategoricals}(\boldsymbol{x}', m_c)$ $\qquad \triangleright$ project $m_c$ indices in $\boldsymbol{x}'$ to categorical space.

---

## C   EXPERIMENTAL DETAILS

### C.1   DATASETS

We use the preprocessed datasets from CARLA library where all categorical features from original datasets have been converted to binary categorical features, for example, place of native country has been converted to US, Non-US, race into white and non-white etc

The Adult data set Becker & Kohavi (1996) originates from the 1994 Census database, consisting of 13 attributes and 48,832 instances. The classification consists of deciding whether an individual has an income greater than 50,000 USD/year. The train split has 36624 examples and test split has 12208 examples. The features sex and race are set as immutable. Categorical features in this dataset are workclass, marital-status, occupation, relationship, race, sex, native-country.

The HELOC dataset (FICO, 2018) contains anonymized information about the Home Equity Line of Credit applications by homeowners in the US, with a binary response indicating whether or not the applicant has even been more than 90 days delinquent for a payment. The dataset consists of 9871 rows and 21 features. The train split has 7403 examples and test split has 2468 examples. All features in this dataset are continuous and mutable.

The COMPAS data set (Angwin et al., 2016) contains data for criminal defendants in Florida, USA. It is used by the jurisdiction to score defendant's likelihood of reoffending. The classification task consists of classifying an instance into high risk of recidivism (score). The dataset consists of 6172 rows and 7 features. The train split has 4629 examples and test split has 1543 examples. Immutable features for COMPAS are sex and race. Categorical features are two_year_recid, c_charge_degree, race and sex.

| Dataset | #Features | #Categoricals | #Immutables | #Positives | #Negatives |
|---------|-----------|---------------|-------------|------------|------------|
| Adult Income | 13 | 7 | 2 | 8,742 | 27,877 |
| COMPAS | 7 | 4 | 2 | 3,764 | 865 |
| HELOC | 21 | 0 | 0 | 3,548 | 3,855 |

Table 5: Training Data Statistics

We normalise each feature in every dataset to range $[0, 1]$.

## C.2 HYPERPARAMETERS

### C.2.1 TRUE CLASSIFIERS

We train a RandomForestClassifier calibrated with CalibratedClassifierCV using isotonic calibration. We generate binary labels by sampling from the predictive probabilities for each input. We also provide sample weights to the `.fit` function. The data is split into train and test after getting true classifiers and below we report various metrics on this test split. This input and its corresponding generated label will be utilized as training data for the ANN classifiers.

| Dataset | Accuracy | ROC AUC | Precision | Recall | F1-Score | Briar Score |
|---------|----------|---------|-----------|--------|----------|-------------|
| Adult Income | 94.36 | 0.9996 | 0.90 | 0.96 | 0.93 | 0.05 |
| COMPAS | 85.74 | 0.9886 | 0.78 | 0.91 | 0.81 | 0.10 |
| HELOC | 100.00 | 1.0000 | 1.00 | 1.00 | 1.00 | 0.02 |

Table 6: Performance Metrics for the trained classifiers. Precision, Recall and F1-score are macro averaged. Briar Score determines the quality of calibration – lower the better.

### C.2.2 PREDICTIVE MODELS

We use train fully connected ReLU models with 10,10,10 layers using learning rate=0.001 and number of epochs =100, batch size = 64

| Dataset | Accuracy | ROC AUC | Precision | Recall | F1-Score |
|---------|----------|---------|-----------|--------|----------|
| Adult Income | 77.33 | 0.86 | 0.76 | 0.78 | 0.76 |
| COMPAS | 69.60 | 0.75 | 0.68 | 0.69 | 0.68 |
| HELOC | 74.23 | 0.80 | 0.74 | 0.74 | 0.74 |

Table 7: Performance Metrics for the trained classifiers. Precision, Recall and F1-score are macro averaged

### C.2.3 BASELINES

| Method | Hyperparameters | Other Params |
|---|---|---|
| `Wachter` | loss_type: "BCE" 
 binary_cat_features: True | - |
| `GS` | - | - |
| `DiCE` | loss_type: "BCE" 
 binary_cat_features: True | - |
| `ROAR` | delta: 0.01 | - |
| `PROBE` | loss_type: "BCE" 
 invalidation_target: 0.05 
 noise_variance: 0.01 | - |
| `TAP` | n_iter: 1000 
 step_size: 0.01 | Verifier Params: 
 layers: [50,50,50,50] 
 epochs: 2000 
 lr: 0.0001 
 batch_size: 128 |
| `CCHVAE` | n_search_samples: 100 
 step: 0.1 
 max_iter: 1000 | VAE Params: 
 layers: [512, 256, 8] 
 lambda_reg: 0.000001 
 epochs: 100 
 lr: 0.001 
 batch_size: 32 |
| `CRUDS` | lambda_param: 0.001 
 optimizer: "RMSprop" 
 lr: 0.008 
 max_iter: 2000 | VAE Params: 
 layers: [512, 256, 8] 
 train: True 
 epochs: 100 
 lr: 0.001 
 batch_size: 32 |

Table 8: Hyperparameters used for baseline methods

### C.2.4 GENRE

For each feature in the input we have a learnable position embedding which we then pass into the transformer model. We use the same architecture across all the datasets, below we describe the various parameters.

| Parameter | Value |
|---|---|
| Number of Bins | 50 |
| Number of layers in Encoder | 16 |
| Number of layers in Decoder | 16 |
| Embedding Size | 32 |
| Feed Forward Dim | 32 |

Table 9: Architecture specifications for the Transformer Layer used.

All datasets uses learning rate = 1e-4 and batch size 16384 for Adult Income dataset else 2048.

### C.3 EVALUATION

We use entire dataset train+test, with predicted label assigned by the true classifiers and train a class conditional LocalOutlierFactor from sklearn. We use `n_neighbours=5, novelty = True`. Results shown in 2 are averaged over 200 instances from test split.

# D ADDITIONAL EXPERIMENTS

## D.1 COMPARISON WITH BASELINES ALONG WITH STANDARD DEVIATION

| Dataset | Adult Income | | | | COMPAS | | | | HELOC | | | |
|---|---|---|---|---|---|---|---|---|---|---|---|---|
| Metric | Cost | Val | LOF | Score | Cost | Val | LOF | Score | Cost | Val | LOF | Score |
| Wachter | 0.31±0.16 | 0.51±0.50 | 0.76±0.43 | 1.24 | 0.20±0.17 | 0.59±0.49 | 0.23±0.42 | 0.79 | 0.79±0.47 | 0.35±0.48 | 0.97±0.17 | 1.28 |
| GS | 1.82±1.07 | 0.40±0.49 | 0.49±0.50 | 0.75 | 1.20±0.73 | 0.66±0.47 | 0.48±0.50 | 0.97 | 0.58±0.38 | 0.32±0.47 | 0.94±0.25 | 1.23 |
| DICE | 0.22±0.10 | 0.62±0.49 | 0.73±0.45 | 1.33 | 0.19±0.16 | 0.57±0.49 | 0.44±0.50 | 0.99 | 0.49±0.31 | 0.34±0.47 | 0.97±0.16 | 1.29 |
| ROAR | 10.17±2.62 | 0.96±0.18 | 0.01±0.07 | 0.19 | 4.01±2.01 | 0.87±0.34 | 0.01±0.07 | 0.30 | 9.64±6.89 | 0.47±0.50 | 0.17±0.38 | 0.19 |
| TAP | 1.10±0.93 | 0.99±0.10 | 0.67±0.47 | 1.57 | 1.46±0.59 | 0.88±0.32 | 0.70±0.46 | 1.37 | 0.71±0.26 | 0.36±0.48 | 0.53±0.50 | 0.86 |
| PROBE | 33.91±14.70 | 1.00±0.00 | 0.00±0.00 | -1.61 | 5.77±4.63 | 0.82±0.38 | 0.00±0.00 | -0.00 | 5.37±5.12 | 0.69±0.46 | 0.07±0.25 | 0.49 |
| CCHVAE | 2.11±1.07 | 0.00±0.00 | 1.00±0.00 | 0.84 | 3.03±0.87 | 1.00±0.00 | 0.07±0.26 | 0.64 | 3.58±0.63 | 0.04±0.21 | 0.14±0.35 | 0.02 |
| CRUDS | 3.17±1.11 | 1.00±0.00 | 0.96±0.18 | 1.72 | 1.10±0.82 | 0.98±0.12 | 1.00±0.00 | 1.83 | 4.30±2.23 | 1.00±0.00 | 0.57±0.50 | 1.37 |
| GenRe | 0.69±0.30 | 1.00±0.00 | 0.98±0.12 | 1.93 | 0.51±0.18 | 0.99±0.10 | 0.97±0.17 | 1.89 | 2.01±0.59 | 1.00±0.00 | 1.00±0.00 | 1.90 |

Table 10: Comparison with baselines along with standard deviation

## D.2 PERFORMANCE ACROSS DIFFERENT TEMPERATURE $\tau$ AND $\sigma$ SETTINGS

| Dataset | Adult Income | | | | COMPAS | | | | HELOC | | | |
|---|---|---|---|---|---|---|---|---|---|---|---|---|
| $\tau, \sigma$ | Cost | Val | LOF | Score | Cost | Val | LOF | Score | Cost | Val | LOF | Score |
| 5.0, 2e-07 | 0.73±0.32 | 0.99±0.10 | 0.94±0.25 | 1.87 | 0.54±0.21 | 0.99±0.10 | 0.86±0.34 | 1.78 | 2.12±0.62 | 1.00±0.00 | 1.00±0.00 | 1.9 |
| 10.0, 2e-07 | 0.70±0.32 | 0.99±0.07 | 0.97±0.16 | 1.92 | 0.50±0.18 | 1.00±0.00 | 0.96±0.20 | 1.89 | 2.00±0.57 | 1.00±0.00 | 1.00±0.00 | 1.9 |
| 15.0, 2e-07 | 0.68±0.30 | 0.99±0.07 | 0.99±0.10 | 1.93 | 0.50±0.18 | 1.00±0.00 | 0.96±0.20 | 1.89 | 1.95±0.57 | 1.00±0.00 | 1.00±0.00 | 1.91 |
| 5.0, 2e-06 | 0.72±0.30 | 0.97±0.16 | 0.96±0.20 | 1.88 | 0.54±0.19 | 0.99±0.10 | 0.88±0.32 | 1.79 | 2.08±0.58 | 1.00±0.00 | 1.00±0.00 | 1.9 |
| 10.0, 2e-06 | 0.69±0.30 | 0.99±0.07 | 0.99±0.10 | 1.93 | 0.51±0.18 | 0.99±0.07 | 0.97±0.16 | 1.9 | 2.00±0.58 | 1.00±0.00 | 1.00±0.00 | 1.9 |
| 15.0, 2e-06 | 0.68±0.29 | 0.99±0.10 | 0.98±0.14 | 1.92 | 0.50±0.18 | 1.00±0.00 | 0.96±0.18 | 1.89 | 1.96±0.57 | 1.00±0.00 | 1.00±0.00 | 1.91 |
| 5.0, 2e-05 | 0.73±0.33 | 0.99±0.10 | 0.97±0.17 | 1.9 | 0.53±0.20 | 0.99±0.07 | 0.89±0.31 | 1.81 | 2.08±0.61 | 1.00±0.00 | 1.00±0.00 | 1.9 |
| 10.0, 2e-05 | 0.69±0.30 | 0.98±0.12 | 0.97±0.16 | 1.91 | 0.51±0.18 | 0.99±0.07 | 0.96±0.20 | 1.88 | 2.00±0.60 | 1.00±0.00 | 1.00±0.00 | 1.9 |
| 15.0, 2e-05 | 0.68±0.30 | 0.99±0.10 | 0.98±0.14 | 1.92 | 0.50±0.18 | 1.00±0.00 | 0.97±0.17 | 1.9 | 1.95±0.56 | 1.00±0.00 | 1.00±0.00 | 1.91 |
| 5.0, 2e-4 | 0.73±0.34 | 0.99±0.07 | 0.94±0.24 | 1.88 | 0.53±0.19 | 0.99±0.10 | 0.88±0.33 | 1.79 | 2.10±0.60 | 1.00±0.00 | 1.00±0.00 | 1.9 |
| 10.0, 2e-4 | 0.70±0.32 | 0.99±0.07 | 0.97±0.16 | 1.92 | 0.51±0.18 | 1.00±0.00 | 0.96±0.20 | 1.89 | 2.00±0.58 | 1.00±0.00 | 1.00±0.00 | 1.9 |
| 15.0, 2e-4 | 0.68±0.29 | 1.00±0.00 | 0.98±0.14 | 1.93 | 0.50±0.18 | 1.00±0.00 | 0.97±0.16 | 1.9 | 1.94±0.55 | 1.00±0.00 | 1.00±0.00 | 1.91 |
| 5.0, 2e-3 | 0.74±0.37 | 0.97±0.17 | 0.93±0.26 | 1.84 | 0.54±0.20 | 0.99±0.07 | 0.84±0.36 | 1.76 | 2.12±0.60 | 1.00±0.00 | 1.00±0.00 | 1.9 |
| 10.0, 2e-3 | 0.69±0.30 | 1.00±0.00 | 0.99±0.10 | 1.94 | 0.51±0.18 | 1.00±0.00 | 0.95±0.21 | 1.88 | 1.99±0.56 | 1.00±0.00 | 1.00±0.00 | 1.91 |
| 15.0, 2e-3 | 0.69±0.32 | 0.99±0.10 | 0.98±0.14 | 1.92 | 0.50±0.18 | 1.00±0.00 | 0.95±0.21 | 1.88 | 1.94±0.56 | 1.00±0.00 | 1.00±0.00 | 1.91 |
| 5.0, 2e-2 | 0.75±0.32 | 0.97±0.16 | 0.92±0.28 | 1.83 | 0.56±0.19 | 0.98±0.12 | 0.77±0.42 | 1.67 | 2.26±0.62 | 0.98±0.12 | 1.00±0.00 | 1.88 |
| 10.0, 2e-2 | 0.72±0.29 | 0.94±0.23 | 0.93±0.26 | 1.82 | 0.55±0.18 | 0.98±0.12 | 0.79±0.41 | 1.69 | 2.22±0.59 | 0.98±0.14 | 1.00±0.00 | 1.87 |
| 15.0, 2e-2 | 0.73±0.30 | 0.94±0.23 | 0.92±0.28 | 1.8 | 0.55±0.18 | 0.99±0.07 | 0.78±0.41 | 1.7 | 2.18±0.60 | 0.98±0.14 | 1.00±0.00 | 1.88 |

Table 11: Performance of GenRe across $\tau\{5.0, 10.0, 15.0\}$ and $\sigma\{2e-7, 2e-6, \ldots, 2e-2\}$

## D.3 COMPARISON WITH NEAREST NEIGHBOR SEARCH ALONG WITH STANDARD DEVIATION

| Dataset | Adult Income | | | | COMPAS | | | | HELOC | | | |
|---|---|---|---|---|---|---|---|---|---|---|---|---|
| Metric | Cost | VaL | LOF | Score | Cost | VaL | LOF | Score | Cost | VaL | LOF | Score |
| NNR | 0.36±0.20 | 0.48±0.50 | 0.89±0.32 | 1.34 | 0.17±0.16 | 0.86±0.35 | 0.88±0.33 | 1.71 | 1.22±0.41 | 0.58±0.49 | 1.00±0.00 | 1.53 |
| NNR($\gamma = 0.7$) | 0.46±0.24 | 0.72±0.45 | 0.84±0.36 | 1.53 | 0.43±0.19 | 0.99±0.10 | 0.97±0.16 | **1.9** | 1.52±0.50 | 0.69±0.46 | 1.00±0.00 | 1.62 |
| NNR($\gamma = 0.7, y = 1$) | 0.47±0.24 | 0.85±0.35 | 0.89±0.31 | 1.71 | 0.45±0.21 | 0.99±0.10 | 0.97±0.16 | **1.9** | 1.57±0.53 | 0.96±0.20 | 0.99±0.07 | 1.88 |
| GenRe($\lambda = 0.5$) | 2.28±0.86 | 0.99±0.10 | 0.99±0.10 | 1.8 | 1.95±0.69 | 1.00±0.00 | 0.92±0.27 | 1.64 | 2.64±0.71 | 1.00±0.00 | 1.00±0.00 | 1.87 |
| GenRe($\lambda = 1.0$) | 1.66±0.91 | 0.92±0.28 | 0.99±0.10 | 1.78 | 1.34±0.58 | 1.00±0.00 | 0.95±0.21 | 1.76 | 2.60±0.68 | 1.00±0.00 | 1.00±0.00 | 1.88 |
| GenRe($\lambda = 2.5$) | 0.81±0.39 | 0.89±0.31 | 1.00±0.00 | 1.83 | 1.02±0.58 | 0.98±0.12 | 0.98±0.12 | 1.82 | 2.43±0.68 | 1.00±0.00 | 1.00±0.00 | 1.88 |
| GenRe($\lambda = 5.0$) | 0.69±0.30 | 1.00±0.00 | 0.98±0.12 | **1.93** | 0.51±0.18 | 0.99±0.10 | 0.97±0.17 | **1.89** | 2.01±0.59 | 1.00±0.00 | 1.00±0.00 | **1.9** |
| GenRe($\lambda = 10.0$) | 0.66±0.28 | 0.98±0.14 | 0.99±0.07 | **1.92** | 0.48±0.19 | 0.95±0.21 | 0.86±0.35 | 1.75 | 1.84±0.55 | 1.00±0.00 | 1.00±0.00 | **1.91** |

Table 12: **Best** is highlighted in bold and **second best** is highlighted bold blue

## D.4 DETAILS ON EXPERIMENT WITH TABSYN AND SVDD

SVDD requires a reward function to guide diffusion model such as TabSyn towards a required distribution. For a given diffusion model which can sample from $P(x)$, SVDD allows us to sample from a distribution $P'(x)$ such that,

$$P'(x) \propto P(x) \exp\left(\lambda \cdot r(x)\right) \tag{10}$$

where $r(x)$ is a reward function which models how desirable a generated example is. To sample from 5, we first train a diffusion model on the data which satisfies our constraints and then for each

$x$ we define the reward function as,

$$r(\boldsymbol{x}') = -\ell_2(\boldsymbol{x}', \boldsymbol{x}) - 100 \cdot \mathbf{1}[\boldsymbol{x}'_i \neq \boldsymbol{x}_i]$$

$\boldsymbol{x}_i$ refers to the subset of features which are immutable. We add a penalty of 100 to the usual cost whenever immutability constraints are violated.

| Dataset | Adult Income | | | | COMPAS | | | | HELOC | | | |
|---|---|---|---|---|---|---|---|---|---|---|---|---|
| Metric | Cost | Val | LOF | Score | Cost | Val | LOF | Score | Cost | Val | LOF | Score |
| TabSyn+SVDD($\lambda = 5.0$) | 3.11±1.44 | 0.99±0.10 | 0.83±0.38 | 1.58 | 2.18±0.77 | 1.00±0.00 | 0.69±0.46 | 1.37 | 3.14±0.84 | 0.98±0.14 | 0.97±0.17 | 1.80 |
| TabSyn+SVDD($\lambda = 10.0$) | 3.10±1.39 | 1.00±0.00 | 0.87±0.34 | 1.63 | 2.20±0.82 | 1.00±0.00 | 0.79±0.41 | 1.48 | 2.85±0.78 | 0.99±0.10 | 0.99±0.10 | 1.84 |
| TabSyn+$Q$ | 0.86±0.39 | 1.00±0.00 | 0.79±0.41 | 1.72 | 0.78±0.44 | 0.99±0.07 | 0.81±0.39 | 1.69 | 2.41±0.65 | 0.99±0.10 | 0.99±0.07 | 1.87 |
| GenRe | 0.69±0.30 | 1.00±0.00 | 0.98±0.12 | **1.93** | 0.51±0.18 | 0.99±0.10 | 0.97±0.17 | **1.89** | 2.01±0.59 | 1.00±0.00 | 1.00±0.00 | **1.90** |

## D.5 PERFORMANCE OF GENRE WITH THE AMOUNT OF DATA

To assess scalability we conducted an experiment in which we train the model on various subsets of data, below we report the results on Adults Income dataset – largest dataset used in this paper.

| Fraction of Data | Time | Cost | Val | LOF | Score |
|---|---|---|---|---|---|
| 0.1 | 798.94 | 1.09±0.66 | 0.92±0.27 | 0.85±0.35 | 1.69 |
| 0.2 | 1454.15 | 1.23±0.63 | 1.00±0.00 | 0.99±0.07 | 1.90 |
| 0.4 | 2768.06 | 0.94±0.51 | 0.97±0.17 | 0.92±0.27 | 1.82 |
| 0.8 | 5219.66 | 0.67±0.28 | 0.92±0.27 | 0.98±0.14 | 1.85 |
| 1.0 | 8464.71 | 0.69±0.30 | 1.00±0.00 | 0.98±0.12 | 1.93 |

Table 13: Performance metrics for different fractions of data.

