# OpenReview forum: "From Search to Sampling: Generative Models for Robust Algorithmic Recourse"
_ICLR.cc/2025/Conference — ICLR 2025 Poster_

### Official Review · Reviewer_SyMJ · 2024-10-29

**Soundness:** 3
**Presentation:** 3
**Contribution:** 2
**Rating:** 6
**Confidence:** 2

**Summary:**

This paper addresses the problem of algorithmic recourse for individuals adversely impacted by automated model decisions. It proposes a generative model based method to finish recourse task by directly sampling without optimization at inference. The main goal is to balance three objectives: proximity to the original profile to minimize cost, plausibility to ensure realistic changes, and validity to achieve the desired result. Experimental results shows its effectiveness.

**Strengths:**

* This paper provides a thorough analysis of the limitations in current algorithmic recourse methods, highlighting the challenges of separately optimizing proximity, plausibility, and validity, which often leads to suboptimal results. This establishes a solid foundation and motivation for the proposed method.
* The GenRe method is intuitive and easy to implement and consistently providing more effective trade-offs among different factors.

**Weaknesses:**

1. **Advantage beyond Nearest neighbor**: The biggest issue is the neccessity of involving a complicated auto-regressive method. Essentially, this method uses a model to learn a smoothly-mixed version of nearest-neighbor search in positive samples. It lacks detailed analysis and ablation study for its advantage beyond simple KNN. Why should we bother to train a huge transformer instead of just finding the nearest valid sample and imitate it? I believe there are subtleties here because it requires some generalization stuff. But all these aspects are absent in the paper.
2. **Interpretability Issue**: While GenRe is effective in balancing recourse objectives, the generative approach may lead to less interpretable outputs, especially for stakeholders who require clear, actionable insights. The model could benefit from an analysis of interpretability or user-friendliness of the generated recourse.

**Questions:**

See weakness.

Also, after reading the original definition of LOF, it is higher when the data point is more abnormal (out of distribution). Why is there a upper arrow in table2?

---

> ### Author Response · Authors · 2024-11-21
> **Rebuttal Response**
>
> Thanks for reviewing our work and constructive feedback.
> ### Q1: Advantage beyond the Nearest Neighbor
>
> Yes, you raised an important point.  We present below results of additional experiments to justify the need for additional smoothing offered by the trained model. These experiments demonstrate that the nearest neighbor methods ignore density and thus can succumb to outliers.  As described in section 2, we want instances that are both near neighbors (small cost) and plausible.  It is difficult to trade off the two terms without a smoothing model.  We compare with three variants of nearest neighbor methods:
>
> 1. NNR: We return the nearest example in the training data which receives the desired label from h, i.e $h(x_i) > 0.5$
> 2. NNR($\gamma=0.7$): We increase the threshold to 0.7 instead, $h(x_i) > \gamma$,
> 3. NNR($\gamma=0.7,y=1$): We return the nearest neighbor from $\mathcal{D}^+$ as described in section . $h(x_i) > \gamma, y_i=1$
>
> We note that the last variant corresponds to our method if we set $\lambda  = \infty$ in eq 1 but without learning any model.
> | Dataset                   | Adult Income |            |            |               | COMPAS     |            |            |              | HELOC      |            |            |               |
> | :------------------------ | :----------- | :--------- | :--------- | ------------: | :--------- | :--------- | :--------- | -----------: | :--------- | :--------- | :--------- | ------------: |
> | Metric                    | Cost         | VAL        | LOF        | Score         | Cost       | VAL        | LOF        | Score        | Cost       | VAL        | LOF        | Score         |
> | NNR                       | 0\.36±0.20   | 0\.48±0.50 | 0\.89±0.32 | 1\.34         | 0\.17±0.16 | 0\.86±0.35 | 0\.88±0.33 | 1\.71        | 1\.22±0.41 | 0\.58±0.49 | 1\.00±0.00 | 1\.53         |
> | NNR$(\\gamma=0\.7)$       | 0\.46±0.24   | 0\.72±0.45 | 0\.84±0.36 | 1\.53         | 0\.43±0.19 | 0\.99±0.10 | 0\.97±0.16 | **1\.9** | 1\.52±0.50 | 0\.69±0.46 | 1\.00±0.00 | 1\.62         |
> | NNR$(\\gamma=0\.7, y=1)$  | 0\.47±0.24   | 0\.85±0.35 | 0\.89±0.31 | 1\.71         | 0\.45±0.21 | 0\.99±0.10 | 0\.97±0.16 | **1\.9** | 1\.57±0.53 | 0\.96±0.20 | 0\.99±0.07 | 1\.88         |
> | GenRe($\\lambda$ = 0\.5)  | 2\.28±0.86   | 0\.99±0.10 | 0\.99±0.10 | 1\.8          | 1\.95±0.69 | 1\.00±0.00 | 0\.92±0.27 | 1\.64        | 2\.64±0.71 | 1\.00±0.00 | 1\.00±0.00 | 1\.87         |
> | GenRe($\\lambda$ = 1\.0)  | 1\.66±0.91   | 0\.92±0.28 | 0\.99±0.10 | 1\.78         | 1\.34±0.58 | 1\.00±0.00 | 0\.95±0.21 | 1\.76        | 2\.60±0.68 | 1\.00±0.00 | 1\.00±0.00 | 1\.88         |
> | GenRe($\\lambda$ = 2\.5)  | 0\.81±0.39   | 0\.89±0.31 | 1\.00±0.00 | 1\.83         | 1\.02±0.58 | 0\.98±0.12 | 0\.98±0.12 | 1\.82        | 2\.43±0.68 | 1\.00±0.00 | 1\.00±0.00 | 1\.88         |
> | GenRe($\\lambda$ = 5\.0)  | 0\.69±0.30   | 1\.00±0.00 | 0\.98±0.12 | **1.93** | 0\.51±0.18 | 0\.99±0.10 | 0\.97±0.17 | *1.89*    | 2\.01±0.59 | 1\.00±0.00 | 1\.00±0.00 | *1\.9*      |
> | GenRe($\\lambda$ = 10\.0) | 0\.66±0.28   | 0\.98±0.14 | 0\.99±0.07 | *1\.92*     | 0\.48±0.19 | 0\.95±0.21 | 0\.86±0.35 | 1\.75        | 1\.84±0.55 | 1\.00±0.00 | 1\.00±0.00 | **1\.91** |
>
> Best is highlighted as **bold** and second best is *italicized*.
>
> We see that NNR has poorer validity and LOF on Adults Income  and HELOC datasets. Upon inspection we also note that there is a tendency to return the same recourse instances for different negative instances. GenRe, while learning a smoothly-mixed version
>
> ### Q2: Interpretability Issue
>
> As noted by previous work(Looveren(2021)), it is difficult to define interpretability in practice and they propose that recourse/explanations are interpretable if they are close to the desired data distribution. In our work, we return recourse instances that are plausible in the positive class, and thus are expected to be more interpretable than those returned by the search-based methods where plausibility is not easily enforced.
>
> ### Clarification regarding LOF
> Yes, by definition, LOF is higher for an outlier point but in the sklearn API, `LocalOutlierFactor` predict function returns  -1 refers to outliers and 1 refers to inlier. We normalize it to 0,1 - effectively making the reported metric as a fraction of recourse instances classified as inliers.
>
> [Looveren 2021] Van Looveren, A., & Klaise, J. (2021, September). Interpretable counterfactual explanations guided by prototypes. In Joint European Conference on Machine Learning and Knowledge Discovery in Databases (pp. 650-665). Cham: Springer International Publishing.

---

> > ### Author Response · Authors · 2024-12-02
> > **Gentle Reminder**
> >
> > Dear Reviewer SyMJ,
> >
> > Thank you once again for your insightful feedback. We greatly appreciate the time and effort you have devoted to reviewing our work.
> >
> > As the ICLR discussion phase approaches its conclusion on December 2 (AoE), we wanted to ensure that our responses have sufficiently addressed your concerns. If there are any remaining questions or areas requiring further clarification, please let us know—we would be happy to provide additional details.

---

> ### Author Response · Authors · 2024-11-24
> **Requesting feedback on our rebuttal**
>
> Dear Reviewer SyMJ,
>
> Thank you for your constructive feedback on our paper. With the ICLR public discussion phase ending shortly, we wanted to confirm whether our responses have sufficiently addressed your concerns. If there are any remaining issues, we would be happy to provide additional clarifications.
>
> Thank you very much for your time!

---

### Official Review · Reviewer_sL8f · 2024-11-01

**Soundness:** 3
**Presentation:** 2
**Contribution:** 2
**Rating:** 6
**Confidence:** 3

**Summary:**

The paper introduces **GenRe**, a generative model that provides actionable, realistic recourse for individuals affected by automated decisions. By jointly optimizing for **proximity** (low cost), **plausibility** (realistic changes), and **validity** (desired outcomes), GenRe outperforms traditional methods that address these goals separately. Rather than using gradient-based search, GenRe generates recommendations through efficient sampling, offering robust, balanced recourse. The model's code is publicly available to support further research and applications.

**Strengths:**

GenRe effectively integrates proximity, plausibility, and validity into a single generative model, addressing the core challenges of algorithmic recourse in a unified way. This contrasts with traditional methods that optimize these objectives separately, offering a more balanced solution.

By using forward sampling instead of gradient-based search, GenRe reduces computational costs during inference. This makes the model more scalable and applicable to real-world scenarios where speed is crucial.

The paper demonstrates that GenRe achieves better trade-offs between cost, plausibility, and validity compared to existing methods, providing evidence of its effectiveness across multiple datasets and scenarios.

The authors have made their code available, promoting transparency and enabling other researchers to build on their work, which helps advance the field of algorithmic recourse.

**Weaknesses:**

While GenRe shows improvements in the tested scenarios, its generalizability to highly diverse datasets or domains with different types of constraints might require additional adjustments or fine-tuning.

The paper acknowledges the challenge of synthesizing recourse supervision for training, which may introduce assumptions or heuristics that could impact the model's robustness in practice.

The success of GenRe heavily depends on the quality of the generative model and the training data. If the data is biased or limited, the recourse recommendations might be less effective or skewed.

**Questions:**

Are there particular large-scale use cases you envision as especially suited for GenRe's capabilities? Can you implement GenRe in a modern generative model, such as a pretrained one?

How does it handle biased or limited datasets?

As GenRe integrates multiple complex objectives, how do you ensure interpretability in the generated recommendations?

---

> ### Author Response · Authors · 2024-11-21
> **Rebuttal Response**
>
> ### Q1.a: Dataset Diversity
> We have tested our method on diverse standard benchmark datasets widely used in recourse literature. Adults Income has 14 features, COMPAS has 7, HELOC has 21 features. We provide more details about the datasets in Appendix B.1
>
> | Dataset       | #Features | #Categoricals | #Immutables | #Positives | #Negatives |
> |---------------|-----------|---------------|-------------|------------|------------|
> | Adult Income  | 13        | 7             | 2           | 8,742      | 27,877     |
> | COMPAS        | 7         | 4             | 2           | 3,764      | 865        |
> | HELOC         | 21        | 0             | 0           | 3,548      | 3,855      |
>
> - We have diversity in terms of class imbalance: Adults Income, COMPAS are imbalanced and HELOC is almost perfectly balanced
> - Both Adults Income and COMPAS have a mix of categorical and continuous variables.
>
> ### Q1.b: Different types of Constraints
> We use $\ell_1$ norm as the cost function, which encourages the output to be sparse and we also account for constraints such as immutable attributes like age and gender. We also note that this is one particular choice, different cost function and more constraints can be easily incorporated in our framework
>
> ### Q2: GenRe in a pretrained model
>
> We can use pre trained diffusion models to sample from the target distribution using a guidance method during inference. We ran an experiment using currently the best performing diffusion model for tabular data Tabsyn (Zhang(2024)) and constrained it with a state-of-the-art derivative-free guidance method SVDD(Li(2024)).
>
> | **Dataset**               | **Adult Income**    |            |            |           | **COMPAS**    |            |            |           | **HELOC**    |            |            |           |
> | :------------------------ | :------------------ | :--------- | :--------- | :-------- | :------------ | :--------- | :--------- | :-------- | :----------- | :--------- | :--------- | :-------- |
> | **Metric**                | **Cost**            | **VaL**    | **LOF**    | **Score** | **Cost**      | **VaL**    | **LOF**    | **Score** | **Cost**     | **VaL**    | **LOF**    | **Score** |
> | TabSyn+SVDD($\lambda=5.0$)  | 3\.11±1.44          | 0\.99±0.10 | 0\.83±0.38 | 1\.58     | 2\.18±0.77    | 1\.00±0.00 | 0\.69±0.46 | 1\.37     | 3\.14±0.8    | 0\.98±0.14 | 0\.97±0.17 | 1\.8      |
> | TabSyn+SVDD($\lambda=10.0$) | 3\.10±1.39          | 1\.00±0.00 | 0\.87±0.34 | 1\.63     | 2\.20±0.82    | 1\.00±0.00 | 0\.79±0.41 | 1\.48     | 2\.85±0.78   | 0\.99±0.10 | 0\.99±0.10 | 1\.84     |
> | GenRe($\lambda = 5.0$)      | 0\.69±0.30          | 1\.00±0.00 | 0\.98±0.12 | **1\.93** | 0\.51±0.18    | 0\.99±0.10 | 0\.97±0.17 | **1\.89** | 2\.01±0.59   | 1\.00±0.00 | 1\.00±0.00 | **1\.9**  |
>
> ### Q3: How does it handle biased or limited datasets?
>
> Our sampling strategy ensures that we only show examples from the desired recourse distribution however imbalanced the label distribution is. As observed from results in Table 2, our method performs well on imbalanced datasets such as  Adults Income and COMPAS
>
> ### Q4: Interpretability
> As noted by previous work(Looveren(2021)), it is difficult to define interpretability in practice and they propose that recourse/explanations are interpretable if they are close to the desired data distribution. In our work, we return recourse instances that are plausible in the positive class, and thus are expected to be more interpretable than those returned by the search-based methods where plausibility is not easily enforced. LOF metric reported in our paper serves as a measure to the notion of interpretablity by Looveren(2021)
>
> ---
>
> [Looveren 2021] Van Looveren, A., & Klaise, J. (2021, September). Interpretable counterfactual explanations guided by prototypes. In Joint European Conference on Machine Learning and Knowledge Discovery in Databases (pp. 650-665). Cham: Springer International Publishing.
>
> [Li(2024)] Li, Xiner, Yulai Zhao, Chenyu Wang, Gabriele Scalia, Gokcen Eraslan, Surag Nair, Tommaso Biancalani et al. "Derivative-free guidance in continuous and discrete diffusion models with soft value-based decoding." arXiv preprint arXiv:2408.08252 (2024).
>
> [Zhang(2024)] Zhang, H., Zhang, J., Srinivasan, B., Shen, Z., Qin, X., Faloutsos, C., Rangwala, H., & Karypis, G. (2024). Mixed-Type Tabular Data Synthesis with Score-based Diffusion in Latent Space. In The twelfth International Conference on Learning Representations.

---

> ### Author Response · Authors · 2024-11-24
> **Requesting feedback on our rebuttal**
>
> Dear Reviewer sL8f,
>
> Thank you for your constructive feedback on our paper. With the ICLR public discussion phase ending shortly, we wanted to confirm whether our responses have sufficiently addressed your concerns. If there are any remaining issues, we would be happy to provide additional clarifications.
>
> Thank you very much for your time!

---

> > ### Comment · Reviewer_sL8f · 2024-11-24
> >
> > Thanks for the response.
> >
> > As I have already given a positive score, I will maintain it.
> >
> > Scalability and interpretability may futher improve the work if possible.

---

> > > ### Author Response · Authors · 2024-11-25
> > > **Clarifications and Further Response**
> > >
> > > Thank you for your feedback. We appreciate your suggestion regarding scalability and interpretability, and we are happy to provide further clarifications.
> > >
> > > ### Scalability
> > >
> > > 1.  **Model parameters comparison**
> > >
> > > |    \#params   | **Adult Income** | **COMPAS** | **HELOC**  |
> > > | :---- | :----| :-----| :---- |
> > > | CRUDS  | 1790260 |1765672|1823044  |
> > > | CCHVAE | 574620 |591956 | 591084 |
> > > | GenRE  | **293608**  | **286564** | **303112** |
> > >
> > > All methods use the same architecture across all the datasets, parameter counts differ due to differences in data dimensionality. It is notable that in GenRe we have almost half the number of parameters compared to other model-based baselines.
> > >
> > > 2. **Training and Inference**:
> > > During inference, our method has a complexity of $O(1)$ whereas most baselines perform search during inference and thus can take longer. Below we provide inference time for one method from each class of methods.
> > >
> > > | **Dataset** | **Adults Income**   |    |           | **COMPAS**      |    |    | **HELOC**       |     |           |
> > > | :--- | :-------: | :----| :----| :----: | :----| :----| :------: | :---- | :--- |
> > > | **Metric**  | **Training(s)**     | **Inference(s)** | **Score** | **Training(s)** | **Inference(s)** | **Score** | **Training(s)** | **Inference(s)** | **Score** |
> > > | DICE  | - | 16\.97  | 1\.33| -    | 51\.29  | 0\.99     | -   | 59\.11  | 1\.29     |
> > > | ROAR  | -  | 1,041\.16  | 0\.19     | -    | 715\.97   | 0\.19     | -    | 673\.41    | 0\.19 |
> > > | CCHVAE  | 776\.09    | 0\.95   | 0\.84     | 86\.1   | 45\.89  | 0\.64 | 168\.56  | 527\.32 | 0\.02 |
> > > | Tabsyn+SVDD | 11513\.9    | 391\.79 | 1\.63     | 5,304\.78 | 250\.65  | 1\.49     | 6,966\.80  | 560\.96  | 1\.84  |
> > > | GenRE   | 8464\.71   | 24\.52  | 1\.93     | 1513\.9 | 7\.49    | 1\.89     | 1913\.43  | 46\.65 | 1\.9 |
> > >
> > > Note that in our implementation of GenRe, we have a fair amount of CPU processing during training when we sample from $Q$(eq 6). This can be eliminated by substituting with sampling routines on gpu available in libraries like torch.
> > >
> > > 3. Additionally to assess scalability we conducted an experiment in which we train the model on various subsets of data, below we report the results on Adults Income dataset -- largest dataset used in this paper.
> > >
> > > | **fraction of data** | **Time** | **Cost**   | **Val**    | **LOF**    | **Score** |
> > > | -------------------: | -------: | ---------: | ---------: | ---------: | --------: |
> > > | 0\.1                 | 798\.94  | 1\.09±0.66 | 0\.92±0.27 | 0\.85±0.35 | 1\.69     |
> > > | 0\.2                 | 1454\.15 | 1\.23±0.63 | 1\.00±0.00 | 0\.99±0.07 | 1\.90     |
> > > | 0\.4                 | 2768\.06 | 0\.94±0.51 | 0\.97±0.17 | 0\.92±0.27 | 1\.82     |
> > > | 0\.8                 | 5219\.66 | 0\.67±0.28 | 0\.92±0.27 | 0\.98±0.14 | 1\.85     |
> > > | 1                    | 8464\.71 | 0\.69±0.30 | 1\.00±0.00 | 0\.98±0.12 | 1\.93     |
> > >
> > > We see that GenRe performs competitively even when only a small fraction of data is available for training
> > >
> > > ### Interpretability:
> > >
> > > We have already highlighted in the original manuscript (as referenced by Looveren et al. (2021)) that interpretability is a challenging concept to define in practice. To address the point further, We posit that,
> > >
> > > 1. Unlike the related area of counterfactual explanations where the examples shown are intended to explain the decision of the classifier, in recourse the goal is only to convince the user that the recourse example is likely to provide the correct labels.
> > > 2. Since we focus on providing examples with greater validity and ensuring that they are realistic and aligned with the positive class, the interpretability of our examples is no less than existing recourse methods.
> > >
> > > We hope this further clarifies our stance on interpretability. If there are any specific criteria for interpretability that you would like us to consider, we would be happy to address them.
> > >
> > > Thank you again for your valuable feedback.

---

### Official Review · Reviewer_m8hr · 2024-11-03

**Soundness:** 3
**Presentation:** 3
**Contribution:** 2
**Rating:** 6
**Confidence:** 4

**Summary:**

The paper introduces GenRe, a generative-based approach to algorithmic recourse that seeks to optimize three core recourse objectives via joint training, including validity, proximity, and plausibility. The authors claim that previous method fail to unify all objectives into the training and instead use a gradient-based inference time searching technique that could lead to unsatisfying performance. GenRe uses an auto-regressive generative model to sample recourse solutions directly, with a hand-craft training objective defined with the cost function and the vanilla data distribution. The authors validate GenRe's performance through comprehensive experiments on synthetic and real-world datasets.

**Strengths:**

- First of all, the paper is well-written and organized, making it easily followed. The problem definition and motivation are presented clearly, allowing readers to understand the contribution.
- Undoubtedly, the problem of recourse via AI is an important and open problem. It is also closely related to conditional generation, and consequently, the technical discussion (if useful) can be apply to a wide range of problems beyond.
- I appreciate that the authors analyze their proposed method on various tasks, helping us to understand the performance of GenRe.

**Weaknesses:**

I think the major weaknesses are a lack of careful motivation of the proposed method, as well as a more direct comparison with methods beyond the field of algorithmic recourse (but can be directly applied). Specifically,

- From the aspect of this specific problem (algorithmic recourse), if I understand correctly, the major improvement in the training process is the goal of Eq (4), which includes all properties that users care about (as well as the empirical distribution $Q(x^+|x)$ for sampling). However, to me, this task is different from sampling over $P(x|y=1)$ only because the additional cost function, $C(x,x')$ should be imposed. As a result, a very clear derivation is to sample from the distribution of $P(x|y=1)\exp^{-\lambda C(x,x')}$. The introduction of $V(x')$ seems to break the probabilistic explanation of the distribution, because $P(y=1|x)$ is naturally contained in $P(x|y=1)$ by the Bayesian law. In addition, the reason for choosing an auto-regressive model for sampling is also left unjustified, as sequential sampling is not a major concern in the problem. More justification (both intuitively and experimentally, and ideally, theoretically) of the technique selection is unavoidable.
- From a more general sense of conditional generation, the problem of cost-aware sampling is not a novel problem and has been studied both in a general scenario and for specific generative models such as AR models or diffusion models. For instance, a direct method if to train a classifier-free diffusion model that allows sampling from $P(X,Y)$, and use a standard guidance technique (such that the $L_2$ norm like the CLIP score) to guide the diffusion process using $e^{-\lambda C(X, X')}$. The proposed training technique for unpair data (Sec 4.1) is, in fact, a very standard processing technique that has been widely used, and the choice of AR model seems to be more strange than another generative model such as VAE and diffusion. Therefore, while I admit that joint training is a new approach (in this field, maybe), the author should compare with the above alternatives to see if the proposed method can outperform existing baselines. One paper for reference: [Learning from Invalid Data: On Constraint Satisfaction in Generative Models]. In addition, [Classifier-Free Diffusion Guidance] is a straight-forward go-to method to compare with.

**Questions:**

No additional question beyond the weaknesses above.

---

> ### Author Response · Authors · 2024-11-21
> **Rebuttal Response**
>
> ### Q1: Necessity of $V(x)$
>
> Yes you are correct, on ignoring the $V(x)$ part, our formulation converts recourse into a cost-aware sampling problem.  Proposing such a formulation is one of our contributions, in contrast to the predominant narrative in the recourse literature of solving an optimization problem during inference to search a recourse instance. The second part of our contribution is harnessing the training data to create instance pairs for implementing the conditional generative model.  These two contributions are agnostic to the specific generative model used.  We were able to engineer the auto-regressive model  for tabular data generation since our motivating applications were all from tabular data. But we also show results with a diffusion model below.
>
> $V(x)$ also has an important role because training labels may be noisy, and this is particularly  true for points that are close to the negative instance that seeks recourse.  Without $V(x)$, $\exp (−λC(x, x'))P (x|y^+)$ may assign greater mass to these noisy labels simply because they are close to $x'$. A good way to visualize this effect is to think of each class distribution as Gaussian with some overlap.  If training examples are sampled from each class independently, the points from $y^+$ class in the overlapping negative region will have low cost. From a Bayesian perspective, we want to sample from $P(x|y, P(y|x) > 0.5)$ instead of $P(x|y)$. Expanding this out we see that it is the normalized form of $P(x|y)V(x)$.
>
> ### Q2: Conditional Generation using Guidance
> We ran an experiment using currently the best performing diffusion model for tabular data Tabsyn (Zhang(2024)) and constrained it with a state-of-the-art derivative-free guidance method SVDD(Li(2024)).  We present the results below.
>  We train diffusion models only on $D^+$ as described in section 4.1.
>
> | **Dataset**               | **Adult Income** |            |            |           | **COMPAS** |            |            |           | **HELOC**  |            |            |           |
> | :------------------------ | :--------------- | :--------- | :--------- | :-------- | :--------- | :--------- | :--------- | :-------- | :--------- | :--------- | :--------- | :-------- |
> | **Metric**                | **Cost**         | **Val**    | **LOF**    | **Score** | **Cost**   | **Val**    | **LOF**    | **Score** | **Cost**   | **Val**    | **LOF**    | **Score** |
> | TabSyn+Q($\lambda=5.0$)     | 0\.86±0.39       | 1.00±0.00 | 0.79±0.41 | 1\.72     | 0\.78±0.44 | 0\.99±0.07 | 0\.81±0.39 | 1\.69     | 2\.41±0.65 | 0\.99±0.10 | 0\.99±0.07 | 1\.87     |
> | TabSyn+SVDD($\lambda=5.0$)  | 3\.11±1.44       | 0.99±0.10 | 0\.83±0.38 | 1\.58     | 2\.18±0.77 | 1\.00±0.00 | 0\.69±0.46 | 1\.37     | 3\.14±0.8  | 0\.98±0.14 | 0\.97±0.17 | 1\.8      |
> | TabSyn+SVDD($\lambda=10.0$) | 3\.10±1.39       | 1.00±0.00 | 0\.87±0.34 | 1\.63     | 2\.20±0.82 | 1\.00±0.00 | 0\.79±0.41 | 1\.48     | 2\.85±0.78 | 0\.99±0.10 | 0\.99±0.10 | 1\.84     |
> | GenRe($\lambda = 5.0$)      | 0\.69±0.30       | 1\.00±0.00 | 0\.98±0.12 | **1\.93** | 0\.51±0.18 | 0\.99±0.10 | 0\.97±0.17 | **1\.89** | 2\.01±0.59 | 1\.00±0.00 | 1\.00±0.00 | **1\.9**  |
>
> As a semi-ideal, time-inefficient baseline, we implement Tabsyn+Q, in which we sample an dataset Dsyn, which is of the same size as the original dataset. The purpose of this baseline is to verify the generation quality of this diffusion model for our task.  For a given negative instance x-, we define the empirical distribution Q on Dsyn as described in Eq. 6. Following our method, we sample 10 recourse instances and return the one with the highest confidence from classifier h.
>
> Tabsyn+Q performs similarly to GenRe on HELOC but performs worse on datasets Adults Income and COMPAS, struggling on the LOF metric. This experiment also sets a skyline that any guidance method should achieve on this diffusion model.
>
> We see that on two of the three datasets the diffusion based generator does not perform well, both in terms of cost and plausibility (low LOF score).  We also observed that, in many instances, the model was not able to produce recourse instances which satisfy the immutability constraints.
> Having said that, the main crux of our contribution is not the specific generative model used, but the idea of converting recourse into a conditional generation task, and our method of creating instance pairs for training the conditional generative model. It may be possible to train a custom conditional diffusion model in place of our auto-regressive model.  However, for these tabular data, auto-regressive models seemed adequate.
>
> ---
> [Li(2024)] Li, Xiner et al. "Derivative-free guidance in continuous and discrete diffusion models with soft value-based decoding." arXiv preprint arXiv:2408.08252 (2024).
>
> [Zhang(2024)] Zhang, H. et al. (2024). Mixed-Type Tabular Data Synthesis with Score-based Diffusion in Latent Space. In The twelfth International Conference on Learning Representations.

---

> ### Author Response · Authors · 2024-11-24
> **Requesting feedback on our rebuttal**
>
> Dear Reviewer m8hr,
>
> Thank you for your constructive feedback on our paper. With the ICLR public discussion phase ending shortly, we wanted to confirm whether our responses have sufficiently addressed your concerns. If there are any remaining issues, we would be happy to provide additional clarifications.
>
> Thank you very much for your time!

---

> > ### Comment · Reviewer_m8hr · 2024-11-25
> >
> > Thanks. After going through your response my major concern of the motivation part is addressed. So, I will raise the score to 6. In terms of comparison with other guidance methods in the broad field, I understand that a more large-scale comparison is not possible given the limited rebuttal period. I encourage the authors to add a more comprehensive and objective comparison in the camera-ready version.

---

> > > ### Author Response · Authors · 2024-11-25
> > > **Thank you for increasing the score**
> > >
> > > Dear Reviewer m8hr,
> > >
> > > We are glad that our rebuttal has addressed your concerns. We sincerely appreciate your thoughtful review and your decision to increase the score. We will try to include a more extensive comparison in the camera-ready version.

---

### Official Review · Reviewer_92zH · 2024-11-04

**Soundness:** 3
**Presentation:** 2
**Contribution:** 3
**Rating:** 6
**Confidence:** 4

**Summary:**

In this work, the authors propose a generative model designed for algorithmic recourse, which transforms a negative sample $\mathbf{x}^-$ into a positive sample $\mathbf{x}^+$. The resulting positive sample must satisfy the following criteria:

- $\mathbf{x}^+$ is a high-quality sample that adheres to the underlying data distribution.
- $\mathbf{x}^+$ is classified with a high probability as belonging to the positive class by a pre-trained classifier.
- $\mathbf{x}^+$ should be close to $\mathbf{x}^-$ in the feature space.

The authors demonstrate that training a single model to simultaneously meet these objectives leads to improved overall performance compared to previous approaches.

**Strengths:**

Some strengths of this work include:

- The development of a single generative model that converts negative examples into positive samples suitable for addressing the algorithmic recourse problem.
- The model enhances the overall performance of recourse mechanisms.
- The authors have anonymously open-sourced their codebase, allowing others to reproduce their approach.

**Weaknesses:**

## Clarity
The notation, definitions, and equations should be carefully rechecked. In the current form of the paper, the following points severely reduce the paper's readability, and the reader has to guess what is most likely implied.

- Quantities are not properly defined.

     1. In Eq. (2), the cost function employed in this work is not defined. What is the cost? Did the authors provide clarifications regarding its final form?
     2. Line 251, $N^+$ is not defined.
     3. Eq (6) what is $\lambda$ values?
     4. Algorithm 1, Line 290: what is $\mathbf{x}$
     5. Algorithm 2: Where is the projectCategoricals defined? What is $m_c$?
     2. Algorithm 2: Line 5, is $k$ sampled? If yes, how?

- Notation Consistency
     1. Are the bin scales the same as bin widths? If yes, please be consistent.
     2. Is $D_1$ the same as $N^+$
     3. Alg. 1: Line 290: $\mathbf{x}$ is not defined.
     3. Alg. 2: Step 1: $\mathbf{x}$ is not defined.
     4. Eq (7) and (8) seem to have different K.
     5. I suggest to limit the notation to $\mathbf{x}^+$ and $\mathbf{x}^-$ whenever possible.


- Are you using a random forest or an MLP for the classifier? If you use both, in which case do you use the first and the second? What is the accuracy of each classifier?

- For the baselines, how are the final checkpoints selected?

- Something is weird; why do you select variance equal to zero (line 376)? This choice makes step 2 of Algorithm 2 redundant.

## Experiments

### Efficiency analysis
- How does the proposed approach differ from prior work regarding the number of parameters, training time, and GPU/time requirements (during training/inference)?

### Ablations
- What is the benefit of using an autoregressive model for generation compared to other architectures? Did the author try a different setup? What is the models' performance for different generative architectures (non-autoregressive)?

## Minor
- Lines 557-560 double entry for the same paper.
- Steps can be added to Alg. 1 similarly to Alg. 2.
- Proofs can be moved to the appendix.

**Questions:**

1. What is the purpose of "TopK" in Line 290? What does "Top" refer to? Is this clarified in the main text? I was unable to find any explanation.
2. How does the model determine which dimensions of $\mathbf{x}^-$ to freeze? Are these dimensions predefined by the dataset? How is the loss function formulated in practice?
3. Have you started training the generative model from scratch, or are you using pre-trained weights?

---

> ### Author Response · Authors · 2024-11-21
> **Rebuttal Response 1/2**
>
> ## Clarity
> Thanks a lot for pointing out notational inconsistencies, we have fixed these in the revised pdf where changes are highlighted in blue.
> Some other clarifications:
> We had provided the cost function details in the experiments sections. We have now also added that information in the problem formulation section.
> We have added a comment explaining `projectCategoricals` function in Algorithm 2. After step 4, we obtain $p_j$, a categorical distribution over bin indices. To sample $k$, we can directly sample from this distribution. In our implementation, we use `torch.multinomial` function to do so.
>
> > Are you using a random forest or an MLP?
>
> We train a random forest classifier with calibrated probabilities on the original dataset to serve as the gold classifier since for recourse tasks we need to establish validity of generated examples.
> From this gold classifier, we sample labels for each example in our dataset. These new sampled labels serve as training data for all downstream models, including the MLP classifier. All methods only access the MLP classifier to generate recourse instances. Previous work on robustness under model shifts also use similar setup [Hamman(2023), Black(2021)]. In section B.2, we provide the performance metrics for both the classifiers.
>
> > For the baselines, how are the final checkpoints selected?
>
> We use complete implementations from CARLA library(https://github.com/carla-recourse), and the final checkpoints used are from the end of the training run after a fixed number of epochs.
>
> > why do you select variance equal to zero (line 376)?
>
> Our method provides a distribution over recourse instances. To compare with the baselines algorithms on the standard benchmark, we used the mode of the sampled bin. This choice corresponds to $\sigma$ = 0. In practice, using our generative approach a user will have the flexibility of inspecting multiple recourse instances by sampling with a non-zero sigma. In the appendix we include results with different temperatures $\tau${5.0,10.0,15.0} and $\sigma${2e-7, 2e-6,..., 2e-2}, below we provide a subset of these results.  From this table, observe that GenRe generates diverse recourse instances (as seen by the varying cost values) while being valid across different $\tau$ and $\sigma$ combinations.
> | **Dataset**| **Adult Income** | |   |     | **COMPAS**    |            |            |           | **HELOC**    |            |            |           |
> | :--- | :---| :---|  :--- | ---: |  :--- |  :--- |  :--- |  ---: |  :--- |  :--- |  :--- |  ---: |
> | $\tau$, $\sigma$ | **Cost**            | **VaL**    | **LOF**    | **Score** | **Cost**      | **VaL**    | **LOF**    | **Score** | **Cost**     | **VaL**    | **LOF**    | **Score** |
> | 5\.0,2e-06  | 0\.72±0.30          | 0\.97±0.16 | 0\.96±0.20 | 1\.88     | 0\.54±0.19    | 0\.99±0.10 | 0\.88±0.32 | 1\.79     | 2\.08±0.58   | 1\.00±0.00 | 1\.00±0.00 | 1\.9      |
> | 10\.0,2e-06   | 0\.69±0.30          | 0\.99±0.07 | 0\.99±0.10 | 1\.93     | 0\.51±0.18    | 0\.99±0.07 | 0\.97±0.16 | 1\.9      | 2\.00±0.58   | 1\.00±0.00 | 1\.00±0.00 | 1\.9      |
> | 15\.0,2e-06 | 0\.68±0.29 | 0\.99±0.10 | 0\.98±0.14 | 1\.92     | 0\.50±0.18    | 1\.00±0.00 | 0\.96±0.18 | 1\.89     | 1\.96±0.57   | 1\.00±0.00 | 1\.00±0.00 | 1\.91     |
> | 5\.0,2e-4     | 0\.73±0.34   | 0\.99±0.07 | 0\.94±0.24 | 1\.88     | 0\.53±0.19    | 0\.99±0.10 | 0\.88±0.33 | 1\.79     | 2\.10±0.60   | 1\.00±0.00 | 1\.00±0.00 | 1\.9      |
> | 10\.0,2e-4       | 0\.70±0.32 | 0\.99±0.07 | 0\.97±0.16 | 1\.92 | 0\.51±0.18    | 1\.00±0.00 | 0\.96±0.20 | 1\.89     | 2\.00±0.58   | 1\.00±0.00 | 1\.00±0.00 | 1\.9      |
> | 15\.0,2e-4       | 0\.68±0.29  | 1\.00±0.00 | 0\.98±0.14 | 1\.93 | 0\.50±0.18    | 1\.00±0.00 | 0\.97±0.16 | 1\.9      | 1\.94±0.55   | 1\.00±0.00 | 1\.00±0.00 | 1\.91     |
> | 5\.0,2e-2  | 0\.75±0.32   | 0\.97±0.16 | 0\.92±0.28 | 1\.83 | 0\.56±0.19    | 0\.98±0.12 | 0\.77±0.42 | 1\.67     | 2\.26±0.62   | 0\.98±0.12 | 1\.00±0.00 | 1\.88  |
> | 10\.0,2e-4  | 0\.72±0.29  | 0\.94±0.23 | 0\.93±0.26 | 1\.82     | 0\.55±0.18    | 0\.98±0.12 | 0\.79±0.41 | 1\.69     | 2\.22±0.59   | 0\.98±0.14 | 1\.00±0.00 | 1\.87  |
> | 15\.0,2e-4| 0\.73±0.30 | 0\.94±0.23 | 0\.92±0.28 | 1\.8      | 0\.55±0.18    | 0\.99±0.07 | 0\.78±0.41 | 1\.7      | 2\.18±0.60   | 0\.98±0.14 | 1\.00±0.00 | 1\.88 |
>
> > Purpose of "TopK"
>
> `TopK` refers to truncation of $Q$ to top $K$ entries. This is done for efficiency reasons. `TopK` takes a categorical distribution over $N$ objects and returns a re-normalised distribution over $K$ objects with highest probabilities.
>
> > Which dimensions of $x^-$ to freeze? How is the loss function formulated?
>
> This is provided by the dataset. For example, in Adults Income dataset, immutable features are race and sex. Loss function is implemented as cross entropy over bin probabilities
>
> > Generative model from scratch, or are you using pre-trained weights?
>
> We train a new generative model from scratch by sampling from $Q$ defined in Eq. 6.

---

> ### Author Response · Authors · 2024-11-21
> **Rebuttal Response 2/2**
>
> ## Experiments
> ### Efficiency Analysis
>  - **Model parameter count comparison:**
> |    \#params   | **Adult Income** | **COMPAS** | **HELOC**  |
> | :--- | :---| :---| :--- |
> | CRUDS  | 1790260 |1765672|1823044  |
> | CCHVAE | 574620 |591956 | 591084 |
> | GenRE  | **293608**  | **286564** | **303112** |
>
> All methods use the same architecture across all the datasets, parameter counts differ due to differences in data dimensionality. It is notable that in GenRe we have almost half the number of parameters compared to other model-based baselines. It might be possible to achieve the same performance with even fewer parameters but we did not tune hyperparameters to optimize for model size.
>
>  - **Training and Inference:** We provide comparisons with same batch and same number of epochs. During inference, our method has a complexity of $\mathcal{O}(1)$ whereas most baselines perform search during inference and thus can take longer. Below we provide inference time for one method from each class of methods.
> | **Dataset** | **Adults Income**   |    |           | **COMPAS**      |    |    | **HELOC**       |     |           |
> | :--- | :-------: | :----| :----| :----: | :----| :----| :------: | :---- | :--- |
> | **Metric**  | **Training(s)**     | **Inference(s)** | **Score** | **Training(s)** | **Inference(s)** | **Score** | **Training(s)** | **Inference(s)** | **Score** |
> | DICE  | - | 16\.97  | 1\.33| -    | 51\.29  | 0\.99     | -   | 59\.11  | 1\.29     |
> | ROAR  | -  | 1,041\.16  | 0\.19     | -    | 715\.97   | 0\.19     | -    | 673\.41    | 0\.19 |
> | CCHVAE  | 776\.09    | 0\.95   | 0\.84     | 86\.1   | 45\.89  | 0\.64 | 168\.56  | 527\.32 | 0\.02 |
> | Tabsyn+SVDD | 11513\.9    | 391\.79 | 1\.63     | 5,304\.78 | 250\.65  | 1\.49     | 6,966\.80  | 560\.96  | 1\.84  |
> | GenRE   | 8464\.71   | 24\.52  | 1\.93     | 1513\.9 | 7\.49    | 1\.89     | 1913\.43  | 46\.65 | 1\.9 |
>
> Note that in our implementation of GenRe, we have a fair amount of CPU processing during training when we sample from $Q$(eq 6). This can be eliminated by substituting with sampling routines on GPU available in libraries like torch.
>
> ### Ablations
> We ran an experiment using currently the best performing diffusion model for tabular data Tabsyn (Zhang(2024)) and constrained it with a state-of-the-art derivative-free guidance method SVDD(Li(2024)).  We present the results below.
>
> | **Dataset** | **Adult Income** |||| **COMPAS** |   |   |  | **HELOC**  |   |    |   |
> | :---- | :--- | :---| :----| :--- | :--- | :--- | :--- | :--- | :--- | :--- | :--- | :--- |
> | **Metric**  | **Cost**         | **Val**    | **LOF**    | **Score** | **Cost**   | **Val**    | **LOF**    | **Score** | **Cost**   | **Val**    | **LOF**    | **Score** |
> | TabSyn+SVDD($\lambda=5.0$)  | 3.11±1.44       | 0.99±0.10 | 0.83±0.38 | 1.58     | 2\.18±0.77 | 1\.00±0.00 | 0\.69±0.46 | 1\.37     | 3\.14±0.8  | 0\.98±0.14 | 0\.97±0.17 | 1\.8      |
> | TabSyn+SVDD($\lambda=10.0$) | 3\.10±1.39       | 1\.00±0.00 | 0\.87±0.34 | 1\.63     | 2\.20±0.82 | 1\.00±0.00 | 0\.79±0.41 | 1\.48     | 2\.85±0.78 | 0\.99±0.10 | 0\.99±0.10 | 1\.84     |
> | GenRe($\lambda = 5.0$)      | 0.69±0.30       | 1.00±0.00 | 0.98±0.12 | **1.93** | 0\.51±0.18 | 0\.99±0.10 | 0\.97±0.17 | **1\.89** | 2\.01±0.59 | 1\.00±0.00 | 1\.00±0.00 | **1\.9**  |
>
> We see that on two of the three datasets the diffusion based generator does not perform well, both in terms of cost and plausibility (low LOF score).  We also observed that, in many instances, the model was not able to produce recourse instances which satisfied the immutable constraints.
>
> Having said that, the main crux of our contribution is not the specific generative model used, but the idea of converting recourse into a conditional generation task, and our method of creating instance pairs for training the conditional generative model.   It may be possible to train a custom conditional diffusion model in place of our auto-regressive model.  However, for these tabular data, auto-regressive models seemed adequate.
>
> ---
> [Hamman(2023)] Faisal Hamman, Erfaun Noorani, Saumitra Mishra, Daniele Magazzeni, and Sanghamitra Dutta. Robust counterfactual explanations for neural networks with probabilistic guarantees. In the International Conference on Machine Learning, pp. 12351–12367. PMLR, 2023.
>
> [Black(2021)] Emily Black, Zifan Wang, Matt Fredrikson, and Anupam Datta. Consistent counterfactuals for deep models. arXiv preprint arXiv:2110.03109, 2021
>
> [Li(2024)] Li, Xiner, Yulai Zhao, Chenyu Wang, Gabriele Scalia, Gokcen Eraslan, Surag Nair, Tommaso Biancalani et al. "Derivative-free guidance in continuous and discrete diffusion models with soft value-based decoding." arXiv preprint arXiv:2408.08252 (2024).
>
> [Zhang(2024)] Zhang, H., Zhang, J., Srinivasan, B., Shen, Z., Qin, X., Faloutsos, C., Rangwala, H., & Karypis, G. (2024). Mixed-Type Tabular Data Synthesis with Score-based Diffusion in Latent Space. In The twelfth International Conference on Learning Representations.

---

> ### Author Response · Authors · 2024-11-24
> **Requesting feedback on our rebuttal**
>
> Dear Reviewer 92zH,
>
> Thank you for your constructive feedback on our paper. With the ICLR public discussion phase ending shortly, we wanted to confirm whether our responses have sufficiently addressed your concerns. If there are any remaining
> issues, we would be happy to provide additional clarifications.
>
> Thank you very much for your time!

---

> > ### Comment · Reviewer_92zH · 2024-11-26
> > **Reviewer Response**
> >
> > The reviewer thanks the authors for their response and for improving their submission based on the reviewers’ comments. Firstly, I want to emphasize that the reviewer believes the proposed method, by adapting all three criteria in the optimization objective, can achieve the best overall performance. However, have the authors considered implementing early stopping during the training of their model and the baselines? Is the performance of the models consistently improving, which is why they opted for a fixed number of epochs for training and used the last updated weights as the checkpoint for evaluation?

---

> > > ### Author Response · Authors · 2024-11-27
> > >
> > > Thank you for reviewing our rebuttal.
> > >
> > > > All three criteria in the optimization objective
> > >
> > > Thank you for acknowledging GenRe’s strength. By modelling all three conflicting criteria in our objective during training, GenRe was able to demonstrate superior performance compared to state-of-the-art baselines.
> > >
> > > > Early Stopping
> > >
> > > Small correction to previous rebuttal response, baseline method TAP which is not from CARLA library does utilize early stopping[6].
> > >
> > > In GenRe, we have also have fixed number of epochs but we keep track of the model which achieves the lowest loss on a 10% subset of the training data[1,2]. Many baselines from the CARLA library do not require model training, and for those that do, we have used their original implementation, which utilise the entire training data to train the model and does not include early stopping[3,4]. The new baseline method TabSyn+SVDD which we implemented during the rebuttal, utilises early stopping during training of the TabSyn model[5].
> > > Below, we provide a summary of the baseline methods.
> > > | Method             | Baseline    | Early stopping? |
> > > | :----------------- | :---------- | :-------------- |
> > > | Needs training     | CRUDS       | No              |
> > > |                    | CCHVAE      | No              |
> > > |                    | TabSyn+SVDD | Yes             |
> > > |                    | TAP         | Yes             |
> > > | Non-Training based | Wachter     | N/A             |
> > > |                    | DICE        | N/A             |
> > > |                    | PROBE       | N/A             |
> > > |                    | GS          | N/A             |
> > > |                    | NNR         | N/A             |
> > >
> > > We kindly ask the reviewer to let us know if they would find it helpful to include CRUDS/CCHVAE baselines with early stopping. If so, we would be happy to extend the CARLA library to support this feature.
> > > Thank you once again for your time and feedback.
> > >
> > > ---
> > > [1] https://anonymous.4open.science/r/GenRe-BD71/scripts/binned.py#L71-L72
> > >
> > > [2] https://anonymous.4open.science/r/GenRe-BD71/scripts/binned.py#L127-L132
> > >
> > > [3] https://github.com/carla-recourse/CARLA/blob/main/carla/recourse_methods/autoencoder/models/csvae.py#L194
> > >
> > > [4] https://github.com/carla-recourse/CARLA/blob/main/carla/recourse_methods/autoencoder/models/vae.py#L107
> > >
> > > [5] https://github.com/amazon-science/tabsyn/blob/main/tabsyn/main.py#L86-L94
> > >
> > > [6] https://github.com/JesseFriedbaum/TAP_code/blob/main/Utils.py#L575-L599

---

> > > > ### Comment · Reviewer_92zH · 2024-11-28
> > > >
> > > > Thank you for your explanation. We believe it would be fairer to select a checkpoint for each model based on the one that achieves the best overall accuracy, considering all or a subset of the metrics used in the paper. It would be great if the authors could include results from the 'best' checkpoints in the final version of their paper.

---

> > > > > ### Author Response · Authors · 2024-11-30
> > > > > **Response to the feedback**
> > > > >
> > > > > Thank you for your feedback.
> > > > >
> > > > > >  “...fairer to select a checkpoint for each model based on the one that achieves the best overall accuracy, considering all or a subset of the metrics used in the paper.”
> > > > >
> > > > > We have considered three metrics in the paper: Validity, LOF and Cost.
> > > > > - Validity measures if the recourse instance receives the desired label from the gold classifier.
> > > > > - LOF measures how close the recourse instance lies on the manifold of data for which the gold classifier assigned the desired label.
> > > > > - Cost - Measures the distance between the negative instance and the recourse instance.
> > > > >
> > > > > Both Validity and LOF rely on the gold classifier, which is not available during training and only used for evaluation. Cost for the recourse task on these models, is not a meaningful metric for selecting the best checkpoint, as doing so can negatively impact the quality of the generative models employed in baselines such as CRUDS and CCHVAE. Note that even GenRe does not utilise any of the above criteria during training, relying on only the training objective.
> > > > >
> > > > > To be more fair, we agree that we should incorporate checkpoint selection and therefore, for both CRUDS and CCHVAE, we have implemented the feature to track the best checkpoint based on their corresponding losses on a 10% holdout subset of the original dataset as is done for GenRe. Below, we present the results before and after incorporating this feature:
> > > > >
> > > > > | **Dataset** | **Adult Income** |            |            |           | **COMPAS** |            |            |           | **HELOC** |            |            |           |
> > > > > | :---------- | :------------------ | :--------- | :--------- | :-------- | :------------ | :--------- | :--------- | :-------- | :----------- | :--------- | :--------- | :-------- |
> > > > > | **Method**  | **Cost**            | **VAL**    | **LOF**    | **Score** | **Cost**      | **VAL**    | **LOF**    | **Score** | **Cost**     | **VAL**    | **LOF**    | **Score** |
> > > > > | CRUDS (last ckpt)      | 3\.17±1.11          | 1\.00±0.00 | 0\.96±0.18 | **1\.72** | 1\.10±0.82    | 0\.98±0.12 | 1\.00±0.00 | **1\.83** | 4\.30±2.23   | 1\.00±0.00 | 0\.57±0.50 | 1\.37     |
> > > > > | CRUDS (best ckpt)   | 4\.06±0.96          | 1\.00±0.00 | 0\.00±0.00 | 0\.69     | 1\.21±0.87    | 1\.00±0.00 | 1\.00±0.00 | **1\.83** | 2\.43±0.61   | 1\.00±0.00 | 1\.00±0.00 | **1\.88** |
> > > > > | CCHVAE (last ckpt)     | 2\.11±1.07          | 0\.00±0.00 | 1\.00±0.00 | 0\.84     | 3\.03±0.87    | 1\.00±0.00 | 0\.07±0.26 | **0\.64** | 3\.58±0.63   | 0\.04±0.21 | 0\.14±0.35 | 0\.02     |
> > > > > | CCHVAE (best ckpt)  | 2\.68±1.08          | 0\.76±0.43 | 1\.00±0.00 | **1\.55** | 1\.65±0.84    | 0\.06±0.23 | 0\.06±0.23 | -0\.13    | 2\.57±0.59   | 0\.95±0.21 | 1\.00±0.00 | **1\.83** |
> > > > >
> > > > > We make the following observations from the above results:
> > > > > 1. Best vs Last Checkpoint (ckpt): a) Overall we observe that selecting best ckpt helped in three out of five cases and worsened the score in other two cases. In case of CRUDS method on COMPAS dataset, we have same score but we see there is an improvement in validity.
> > > > >
> > > > > 2. Even after these changes, these two methods do not outperform GenRE.
> > > > >
> > > > > CRUDS uses Conditional Subspace VAE (CSVAE) which tracks two different losses -- ELBO loss and BCE loss-- and uses two separate optimizers to update the model in each epoch [1,2]. So, it becomes challenging to define one scalar quantity to track the best checkpoint. Original paper on CSVAE (Klys(2018)) mentions qualitative and quantitative validation criteria in Appendix 7.1, but does not specify a concrete validation target and to the best of our knowledge, no official code implementation is available as well. Therefore we only track the ELBO loss, as it measures the generation quality.
> > > > > For more details, please checkout the original training code from CARLA library and the new training code, both of which we provide in the anonymized repository [3]  respectively, in the files `models_original_from_carla.py` and `models_best_ckpt_and_gpu_support.py`
> > > > >
> > > > > Thank you again for your valuable feedback. We hope this comprehensively addresses your concerns and are open to further discussions.
> > > > >
> > > > > ---
> > > > >
> > > > > [Klys(2018)] Klys, J., Snell, J., & Zemel, R. (2018). Learning latent subspaces in variational autoencoders. Advances in neural information processing systems, 31.
> > > > >
> > > > > ---
> > > > >
> > > > > [1] https://github.com/MartinPawel/ProbabilisticallyRobustRecourse/blob/main/carla/recourse_methods/autoencoder/losses/losses.py#L17-L69
> > > > >
> > > > > [2] https://github.com/MartinPawel/ProbabilisticallyRobustRecourse/blob/main/carla/recourse_methods/autoencoder/models.py#L528-L544
> > > > >
> > > > > [3]  https://anonymous.4open.science/r/GenRe-BD71/carla-ckpt/

---

> ### Comment · Reviewer_92zH · 2024-12-01
>
> Dear Authors,
>
> For the choice of standard deviation (Table 11 in updated pdf), it seems like the best results are given for smaller values of $\sigma$ (moving closer to the deterministic case $\sigma=0$). For the additional evaluations, the authors provided different $\sigma$ values, but their range is really small. Even the largest one will result in a variance of $1e-4$.
>
> #### Question 1
>
> How different/diverse can the sampled $\mathbf{x}'$ be for such a small standard deviation or variance? Does your model generate really diverse solutions for such a choice of $\sigma$? Or is it more the case that you find one good $\mathbf{x}'$ for each $\mathbf{x}^-$?
>
> #### Question 2
>
> Can you provide additional evaluations for the above table as you increase the variance or standard deviation until it reaches a value such as 1? Also, can you include a metric to measure the diversity of the sampled $\mathbf{x}'$? How different are these solutions become as we increase the variance? Can we say something about the trade-off between distance from the mean and recourse allocation quality of the sampled $\mathbf{x}'$?

---

> > ### Author Response · Authors · 2024-12-02
> > **On Diversity of Generated Recourse**
> >
> > Thank you once again for engaging us with thoughtful questions.
> >
> > > deterministic case $\\sigma \= 0$....
> >
> > We believe that there is some misunderstanding of the $\\sigma$. We describe the sampling procedure of the autoregressive model (Algorithm 2).
> >
> > We first sample the bin index from the multinomial distribution in step 4\. Then we sample a value within the sampled bin using a Gaussian distribution that has variance $\\sigma^2$. Therefore, the $\\sigma$ that we choose is consequential only for selecting a value within the bin and not for selecting the bin itself.
> > Temperature $\\tau$ controls the sharpness of the multinomial distribution, with higher $\\tau$ biasing it towards the mode, while $\\sigma$ controls how “far” we sample from the bin.
> >
> > So, even setting $\\sigma=0$, doesn’t make the sampling algorithm deterministic, as long as $\\tau$ remains finite.
> >
> > > For the additional evaluations, the authors provided different values, but their range is really small.
> >
> > As we have described in sections 5.1, line 343, the number of bins we consider is 50\. Further since CARLA range-normalizes all features as part of pre-processing, the width of each bin is 1/50 \= 2e-2. This is the standard deviation we use while training the model. Therefore, exploring significantly higher standard deviations is not meaningful.
> > Please refer to \[1\], to see how the number of bins affects the standard deviation used to train the model.
> >
> > > …best results are given for smaller values of $\\sigma$
> >
> > From Table 11, in the updated pdf, we can see that GenRe outperforms all the baseline methods across the entire range of $\\sigma$ and $\\tau$ considered, marginally trailing behind only CRUDS baseline method, only on COMPAS dataset in 7 out of 18 cases. Notably, GenRe consistently outperformed all baselines on higher-dimensional datasets like Adult Income (14 features) and HELOC (21 features), compared to COMPAS, which has only 7 features. This highlights GenRe's scalability compared to baseline methods.
> >
> > ### Question 1:
> >
> > As explained earlier, even $\\sigma=0.0$ will result in diverse recourse instances as long as $\\tau$ is finite. Below we present results from a new experiment where for each negative recourse instance, we sample 10 recourse instances, and calculate average feature wise sample variance. In the table below we provide mean and standard deviation of this quantity over 200 negative instances.
> >
> > | Dataset | Adult Income | COMPAS | HELOC |
> > | ----- | :---: | :---: | :---: |
> > | $\\tau=1.0, \\sigma=0.0$ | 0.0114±0.0090 | 0.0198±0.0172 | 0.0092±0.0031 |
> > | $\\tau=2.0, \\sigma=0.0$ | 0.0030±0.0056 | 0.0052±0.0088 | 0.0057±0.0031 |
> > | $\\tau=5.0, \\sigma=0.0$ | 0.0008±0.0018 | 0.0003±0.0010 | 0.0034±0.0032 |
> > | $\\tau=10.0, \\sigma=0.0$ | 0.0004±0.0012 | 0.0001±0.0002 | 0.0022±0.0027 |
> > | $\\tau=15.0, \\sigma=0.0$ | 0.0002±0.0002 | 0.0001±0.0001 | 0.0014±0.0024 |
> >
> > These results supports that depending upon $\\tau$, GenRe can produce diverse examples even when $\\sigma=0.0$, where recourse instances are more diverse for smaller values of $\\tau$.
> >
> > ### Question 2:
> > We have already justified the choice of the maximum standard deviation. To answer the question regarding change in diversity with respect to $\\sigma$, below we provide results from a similar setup as described in our response to Q1 but over a range of $\\sigma$ values and fixed $\\tau \= 5.0$
> >
> > | Dataset | Adult Income | COMPAS | HELOC |
> > | ----- | ----- | ----- | ----- |
> > | $\\tau=5.0, \\sigma=2e-6$ ($\\sigma \= $ bin width \* 1e-4) | 0.0007±0.0017 | 0.0003±0.0010 | 0.0035±0.0031 |
> > | $\\tau=5.0, \\sigma=2e-4$ ($\\sigma \= $ bin width \* 1e-2) | 0.0008±0.0018 | 0.0003±0.0010 | 0.0033±0.0031 |
> > | $\\tau=5.0, \\sigma=2e-2$ ($\\sigma \= $ bin width) | 0.0012±0.0029 | 0.0004±0.0010 | 0.0056±0.0036 |
> >
> > >  Trade-off between distance from the mean and recourse allocation quality
> >
> > Higher $\\sigma$ allows for more exploration in the space of recourse instances and thus it is possible to select high quality recourse instances \-- from a predefined criteria or according to user preferences, which is one of the strengths of sampling based recourse. However, sampling farther from the probable means will result in less plausible instances.
> >
> > We would greatly appreciate the reviewer's feedback on these findings and are eager to continue the discussion. Thank you once again.
> >
> > ---
> >
> > \[1\] [https://anonymous.4open.science/r/GenRe-BD71/models/binnedpm.py\#L65](https://anonymous.4open.science/r/GenRe-BD71/models/binnedpm.py#L65)

---

> > > ### Comment · Reviewer_92zH · 2024-12-03
> > >
> > > We thank the authors for such a detailed response and for providing statistics for the diversity of the sampled recourse instances $\mathbf{x}'$. The provided analysis and ablations for different temperature and bin width values are insightful and help understand the quality of the GenRe sampled recourse instances.

---

### Comment · Area_Chair_2umj · 2024-11-24

Dear Reviewers,

This is a gentle reminder that the authors have submitted their rebuttal, and the discussion period will conclude on November 26th AoE. To ensure a constructive and meaningful discussion, we kindly ask that you review the rebuttal as soon as possible and verify if your questions and comments have been adequately addressed.

We greatly appreciate your time, effort, and thoughtful contributions to this process.

Best regards,
AC

---

### Author Response · Authors · 2024-12-04
**Rebuttal Summary**

We sincerely thank all the reviewers for their time and thoughtful comments. Below, we highlight the key contributions of our paper and summarize the main points discussed during the rebuttal period.

# Key Contributions:

1. **Algorithmic Recourse as a conditional generation task**: Our primary contribution is framing recourse as a conditional generation task, which offers several advantages over existing methods. The conditional distribution defined in Eq. 5 encapsulates all the criteria we aim to balance, clearly following the ideal recourse objective outlined in Eq. 1\.
2. **Sampling over Search**: During inference, our approach enables the generation of diverse recourse instances through forward sampling, avoiding the brittleness of search-based methods. Gradient search with imperfect estimators, as used by many baseline methods, often performs poorly, as shown in our extensive empirical evaluations. Direct sampling not only improves the quality of the results but also enables faster inference compared to iterative search methods, as demonstrated during discussions.
3. **Training Recipe for Recourse Generative Model**: Next we also demonstrate a practical approach for sampling instance pairs for training  such a recourse model. With Theorem 4.1, we demonstrate that this approach produces a generative model that is consistent with the recourse distribution (Eq. 6\)
   We emphasize that while we demonstrate the effectiveness of this approach using an autoregressive model,  the sampled instance pairs can be used with any model classes which support vector-to-vector conditional generation in a plug and play manner.
4. **Better results compared to baselines**: Through extensive evaluations in Section 5, we showcase the strong performance of GenRe compared to baseline methods, with significant gains on high dimensional datasets. Additionally, vivid illustrations in Fig. 2 provide valuable insights into the strengths and limitations of  various classes of existing recourse methods compared to GenRe.

# Summary of Rebuttal Discussions:

- **Other Realizations of GenRe**: Recent work \[1\] on diffusion models, also allows us to sample from a distribution similar to recourse distribution (Eq. 5\) from pre-trained models. We implemented SVDD \[2\], a SOTA derivative-free-guidance method on TabSyn \[3\], a leading diffusion model for tabular data. In section 5.6, we demonstrate that inference-time guidance does not match the performance of GenRe. This suggests that our approach, which leverages conditional examples during training, outperforms methods that rely on inference-time conditioning.
- **Advantages beyond Nearest Neighbor Search**: We show that searching for recourse instances from near neighbors in the positive distribution mostly returns outlying instances. GenRe generates more representative examples without incurring significantly higher cost. We have included a comparison of such methods with GenRe in section 5.5. Of course, another merit of GenRe is that it can generate novel resource instances.
- **Efficiency**: Despite training with substantially fewer parameters — up to 6x reduction in model size — GenRe outperforms baseline methods across various metrics and inference speed. GenRe’s approach prefers to pay the overheads one-time during training.
- **Diversity**: We conducted a thorough analysis of how the GenRe’s hyperparameters $\\tau$ and $\\sigma$ affect the diversity of generated recourse and the overall performance. As shown in Table 12, while performance varied across different hyperparameter values, GenRe consistently outperformed baselines.
- **Interpretability**: Recourse instances that lie within the data distribution are generally considered as interpretable in the literature. As demonstrated with LOF metric, GenRe produces recourse instances that are inliers in the desired class distribution.
- **Early Stopping for Baselines**: We implemented early stopping for model-based baseline methods in the CARLA library. While it improved their performance in certain cases, they still failed to outperform GenRe across various metrics.

We deeply appreciate the reviewers' thoughtful feedback, suggestions and the engaging discussions, which provided valuable insights and helped us refine our contributions.

---

\[1\] Cheng Lu et. al. 2023\. Contrastive energy prediction for exact energy-guided diffusion sampling in offline reinforcement learning. ICML 23

\[2\] Li, Xiner et al. Derivative-free guidance in continuous and discrete diffusion models with soft value-based decoding. arXiv:2408.08252 (2024).

\[3\] Zhang, H et. al. (2024). Mixed-Type Tabular Data Synthesis with Score-based Diffusion in Latent Space. ICLR 24\.

---

### Meta-Review · Area_Chair_2umj · 2024-12-29

**Metareview:**

This paper proposes GenRe, a generative model approach to algorithmic recourse that aims to help individuals adversely impacted by automated decisions understand how to achieve more favorable outcomes. By jointly optimizing three objectives - proximity to the original profile, plausibility of changes, and validity of outcomes - through a generative model during training rather than during inference, GenRe can produce better recourse recommendations compared to existing methods that optimize these objectives separately. Empirically, GenRe outperforms baseline methods across multiple metrics on both synthetic and real datasets. The reviewers appreciated the problem motivation, thorough empirical validation across diverse datasets, and a novel technical contribution in framing recourse as a conditional generation task rather than an optimization problem. Key weaknesses identified by reviewers include: limited justification for using an autoregressive architecture over simpler nearest-neighbor approaches, lack of comprehensive comparison to broader conditional generation methods beyond the recourse literature, and insufficient analysis of model interpretability and computational efficiency trade-offs. Additionally, while the paper demonstrates improvements over baselines, some reviewers felt the gains may not fully justify the increased model complexity. A future version of the paper could benefit from more detailed ablation studies comparing different model architectures, analysis of computational costs versus performance benefits, and deeper investigation of interpretability aspects that are important for real-world deployment. Overall, the reviewers were in agreement to marginally accept the paper.

**Additional Comments On Reviewer Discussion:**

See above.

---

### Decision · Program_Chairs · 2025-01-22

Accept (Poster)